# Improving Generalization with Approximate Factored Value Functions

**Shagun Sodhani**                                                                    *sshagunsodhani@gmail.com*
*Meta AI*

**Sergey Levine**                                                                     *svlevine@eecs.berkeley.edu*
*University of California, Berkeley*

**Amy Zhang**                                                                         *amyzhang@meta.com*
*Meta AI*
*UT Austin*

**Reviewed on OpenReview:** *https://openreview.net/forum?id=LwEWrrKyja*

## Abstract

Reinforcement learning in general unstructured MDPs presents a challenging learning problem. However, certain MDP structures, such as factorization, are known to simplify the learning problem. This fact is often not useful in complex tasks with high-dimensional state spaces which do not usually exhibit such structure, and even if the structure is present, it is typically unknown. In this work, we instead turn this observation on its head. Instead of developing algorithms for structured MDPs, we propose a representation learning algorithm that approximates an unstructured MDP with one that has factorized structure. We then use these factors as a more convenient representation of the state for downstream learning. The particular structure that we leverage is reward factorization, which defines a more compact class of MDPs that admit factorized value functions. We empirically verify the effectiveness of our approach in terms of faster training (better sample complexity) and robust zero-shot transfer (better generalization) on the ProcGen benchmark and the MiniGrid environments.[1]

## 1 Introduction

Reinforcement Learning problems are often modeled as Markov Decision Processes (MDPs). While the MDP formulation is quite general, it is not always the most optimal. Reinforcement learning in such general, unstructured MDPs faces challenges like sample inefficiency and brittleness (Zhang et al., 2018a;b; Song et al., 2019; Ghosh et al., 2021). To address these challenges, recent works (Zhang et al., 2021a; Sodhani et al., 2022b; Zhang et al., 2021b) have proposed introducing additional assumptions to leverage the structure underlying the given task(s). While it is intuitive to leverage structure when it exists and is available, these approaches may not be helpful when working with complex MDPs with high-dimensional state spaces where that structure often does not exist or is not known.

In this work, we propose to leverage structure in the reverse direction. Instead of developing algorithms for structured MDPs, we propose to map an unstructured MDP to one that is approximate structured, where we can then exploit that structure for improved sample efficiency and generalization. In the scope of this work, we focus on a class of structured MDP that we call a *Factored Reward MDP*, where the state and the rewards are factorized. Formulating an environment as a Factored Reward MDP provides computational benefit as these MDPs are more *compact* than standard MDPs. For example, consider a case where the state space can be factorized into two variables, each taking $m$ unique discrete values. The Factored Reward MDP formulation will use $2^{m+1}$ dimensions, while the standard MDP formulation will use $2^{2 \times m}$ dimensions for

---

[1]Project website: https://sites.google.com/view/factored-representation/home

representing the state. From the cognitive science perspective, the factors can be thought of as reusable *mechanisms* (Battaglia et al., 2018; Bapst et al., 2019; Sanchez-Gonzalez et al., 2020) that can be shared across tasks, thus enabling the learning agent to transfer knowledge across tasks easily and quickly adapt to new tasks.

We also note that the Factored Reward MDPs are not merely a useful theoretical construct. This class of MDPs actually manifests in many real-world scenarios. When working with embedded systems, it is common to concatenate input from different sensors into a single observation vector, resulting in a factorized state space. Suppose the training agent has access to this factorization. In that case, it can use specialized mechanisms for encoding the output of different kind of sensors or can selectively focus on the output of the most relevant sensors (for the task at hand). Similarly, when working on robotics applications, the different joint angles, positions, and velocities are concatenated into a single, large state space Hodge et al. (2021) even though these *factors* could lead to independent rewards. For example, the agent could be rewarded for running at a particular velocity as well as for reaching a specific position. Knowing the underlying factorization could help the learning agent optimize for the overall task.

We show that Factored Reward MDPs exhibit factored value functions which may allow for better generalization to novel combinations of those factors. Existing RL algorithms can be extended to exploit the Factored Reward MDP's structure. Motivated by the advantages of Factored Reward MDPs, we propose a representation learning technique, called **A**pproximately **Fa**ctored **R**epresentations (AFaR), that can map an unstructured MDP into an approximate Factored Reward MDP. We show that this can lead to sample efficiency gains and improved generalization performance even when that mapping is approximate.

**Contributions**: (i) We propose a representation learning technique to map a given MDP into an approximate Factored Reward MDP. ii) We show that the proposed representation learning algorithm can be easily *plugged* into standard actor-critic algorithms and the entire system can be trained end-to-end. (iii) We empirically verify the effectiveness of the proposed algorithm in terms of better sample complexity (faster training) and better generalization (robust zero-shot transfer).

## 2 Problem Setup

We first define the standard assumptions for a Markov decision process (MDP) and introduce a new form of structured MDP we call a Factored Reward MDP in Section 2.1. Next, in Section 2.2, we show that Factored Reward MDP exhibits the useful property of emitting factored value functions. In Section 3, we use these concepts in reverse to design a representation learning approach for transforming standard MDPs into approximate Factored Reward MDPs. In Section 4, we describe how we use this factored representation and the factored value functions to derive a policy.

### 2.1 Preliminaries

A **Markov Decision Process** (MDP) (Bellman, 1957; Puterman, 1995) is defined by a tuple $\langle \mathcal{S}, \mathcal{A}, R, T, \gamma \rangle$, where $\mathcal{S}$ is the set of states, $\mathcal{A}$ is the set of actions, $R : \mathcal{S} \times \mathcal{A} \to \mathbb{R}$ is the reward function, $T : \mathcal{S} \times \mathcal{A} \to Dist(\mathcal{S})$ is the environment transition probability function, and $\gamma \in [0, 1)$ is the discount factor. At each time step, the learning agent perceives a state $s_t \in \mathcal{S}$, takes an action $a_t \in \mathcal{A}$ drawn from a policy $\pi : \mathcal{S} \times \mathcal{A} \to [0, 1]$, and with probability $T(s_{t+1}|s_t, a_t)$ enters next state $s_{t+1}$, receiving a numerical reward $R(s_t, a_t)$ from the environment. The state value function of policy $\pi$ is defined as: $V_\pi(s) = E_\pi[\sum_{t=0}^{\infty} \gamma^t R(s_{t+1})|S_0 = s]$. The optimal state value function $V^*$ is the maximum value function over the class of stationary policies.

A **Factored Reward Markov Decision Process** (Factored Reward MDP) is a special class of MDP where the state and the reward can be factorized into *variables* or *factors*. For example, a state $s_t$ can be factored into $k$ state factors $\{s_t^1, \cdots, s_t^k\}$. The set of possible values that the $i^{th}$ state factor can take are represented as $\mathcal{S}^i$ and the state space $\mathcal{S}$ is the cross product of the value spaces for the individual state factors, i.e. $\mathcal{S} = \times_{i=1}^k \mathcal{S}^i$. We rely on the following assumption:

**Assumption 2.1** (Factored Rewards). *For given full states $s_t \in \mathcal{S}$, action $a_t \in \mathcal{A}$, and $s_t^i$ denoting the $i^{th}$ factor of the state $s_t$, we have $R(s_t, a_t) = \sum_i r^i(s_t^i, a_t)$ where $r^i : \mathcal{S}^i \times \mathcal{A} \to \mathbb{R}$ are the local reward functions.*

Factored Reward MDPs are a useful abstraction for representing high-dimensional MDPs generated by systems with many *weakly* connected components. These MDPs can also be viewed as a relaxed form of the Factored MDPs (Boutilier et al., 1995; 1999; Kearns & Koller, 1999), another class of MDPs that rely on the additional assumption of factored transition function.

## 2.2 Factored Reward MDPs Emit Factored Value Functions

Given the above definition for a Factored Reward MDP, we show that the state value functions and the state-action value functions are also factorized.

**Proposition 2.2.** *Consider a Factored Reward MDP where the state $s_t$ is factorized into $k$ state factors $s_t^1, \cdots, s_t^k$ and the reward function is factorized as $R(s_t, a_t) = \sum_{i=1}^{k} r^i(s_t^i, a_t)$ where $r^i : \mathcal{S}^i \times \mathcal{A} \to \mathbb{R}$ are the local reward functions. The state value function and the state-action value functions for any policy $\pi$ can be written as:*

$$V^\pi(s_t) = \sum_{i=1}^{k} V_i^\pi(s_t^i), \quad Q^\pi(s_t, a_t) = \sum_{i=1}^{k} Q_i^\pi(s_t^i, a_t), \tag{1}$$

*where*

$$V_i^\pi(s_t^i) = \sum_{t'=t}^{\infty} \gamma^{t'} \sum_{s_{t'+1}} T(s_{t'+1}|s_{t'}, a_{t'}) r^i(s_{t'}^i, a_{t'}),$$

$$Q_i^\pi(s_t^i, a_t) = r^i(s_t^i, a_t) + \sum_{t'=t+1}^{\infty} \gamma^{t'} \sum_{s_{t'+1}} T(s_{t'+1}|s_{t'}, a_{t'}) r^i(s_{t'}^i, a_{t'})$$

*are the state value and the state-action value functions for the $i^{th}$ factor, respectively and the actions $a_t$ are obtained using the policy $\pi$.*

The proof is available in Appendix A. Our proposed approach (described in Section 3) does not assume access to factored Q-functions. Instead, it applies these concepts in reverse to learn a representation that transforms standard MDPs into approximate Factored Reward MDPs.

## 3 AFaR: Learning Approximate Factored Representations

In Section 2.2, we showed that Factored Reward MDPs emit factored value functions. We now reverse this observation to propose a representation learning technique that induces such factorization. Specifically, we approximate the value of a state as the sum of the values corresponding to the different factors.[2] In practice, this translates to computing the state-value function for a given state in terms of the sum of the state-value function for the factors of the given state. These factors are learnt using a *Factor Encoder* module, denoted as $\phi$, and is described in detail in Section 3.1. Our hope is that learning the factorized representation could be helpful for improving the generalization performance and sample efficiency of RL agents on environments with factorizable observation spaces. We refer to our proposed approach as **A**pproximately **Fa**ctored **R**epresentations, or AFaR. In Section 4, we discuss leveraging this form of structured representation for downstream control.

### 3.1 Factor Encoder

*Factor Encoder* module, denoted by $\phi$, is an encoder that maps a given input state $s_t$ to a set of $k$ factor representations, denoted as $\{z_t^1, \cdots, z_t^k\})$. Here, $k$ is a hyperparameter. For each factor in the

---

[2]Note that in practice, we use the *mean* operation instead of the *sum* operation to normalize the value function with respect to the number of factors. Given that the number of factors is constant for a given instantiation of the training agent, using mean or sum is equivalent to a multiplicative constant. Using mean, in place of sum, ensures that the magnitude of output of value functions (and training losses) remains of the same order as we vary the number of encoders, there by enabling us to use the same learning rate across experiments.

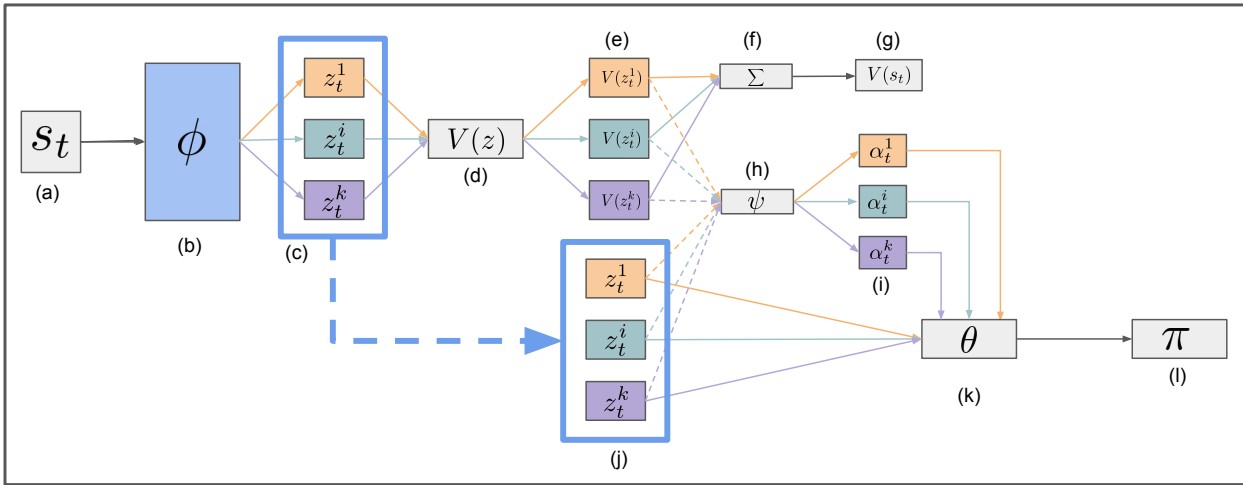

Figure 1: Architecture of the AFaR model: Given an input state $s_t$ (shown as ($a$)), the *Factored Encoder* module $\phi$ (shown as ($b$)) is used to compute the representations for the $k$ factors, denoted as $\{z_t^1, \cdots, z_t^k\}$ (shown as ($c$)). State value is computed for each factor (shown as ($d, e$)) and the value for the given state is obtained by summing over the individual state values (shown as ($f, g$)). The state values, along with factor representations, are used to compute the attention scores $\alpha_t$ (shown as ($h, i$)) which is used for aggregating the factor representations by the *Aggregation Module*, $\theta$, (shown as ($k$)). The aggregated feature representation is used to select the action from the universal policy (shown as ($l$)). Dashed lines indicates the components (or computations) through which gradient does not flow. The blue line and box show that the same factored representations are used in (c) and (j) stages.

factored representation, we compute its corresponding state value, denoted as $V_i = V(z_t^i) \forall i \in \{1, \cdots, k\}$). Following Equation (9), we obtain the state value for the state $s_t$ by summing over the state values corresponding to the individual factors (shown in step $f$ in Figure 1).

The *Factor Encoder* module is instantiated as a neural network, and we overload the notation to denote both the module and the learnable weights (of the encoder) using $\phi$. The *Factor Encoder* module is trained using the critic loss alongside the value function. We note that while we use the subscript $i$ in the definition of the value functions in Equation (9) (e.g. $V_i$), the neural network (instantiating the value functions) are shared across all factors (shown in component $d$ in Figure 1). Using a shared value function helps to ensure that the learned state factors lie in the same representational space.

## 4 Extracting a Policy from Factorized Value Functions

In Section 3, we described our proposed representation learning approach, AFaR, that enables modeling an MDP as an approximate Factored Reward MDP. Since AFaR is a representation learning algorithm, we must pair it with a policy optimization algorithm for end-to-end reinforcement learning. In this section, we describe how to combine AFaR with actor-critic algorithms. We outline a standard actor-critic algorithm in Algorithm 1 and highlight the changes (in red color) for incorporating the AFaR for performing end-to-end reinforcement learning. In Section 5, we evaluate the effectiveness of our proposed approach empirically by testing AFaR in conjunction with three different actor-critic algorithms — Rewarding Impact-Driven Exploration or RIDE (Raileanu & Rocktäschel, 2020), Data-regularized Actor-Critic or DrAC (Raileanu et al., 2020) and Invariant Decoupled Advantage Actor-Critic or IDAAC (Raileanu & Fergus, 2021). For additional details about these algororithms and why we specifically choose them as baseline algorithms, we direct the readers to Appendix E.1.

Previous works have shown that factorized state and value functions alone need not lead to factorized policies (Liberatore, 2002; Allender et al., 2003). As such, we need to combine the factorized representation into a single representation that can be fed into a universal policy[3]. The learning agent aggregates the

---

[3]Here, a universal policy refers to a policy shared across the factors

---

**Algorithm 1** AFaR algorithm.

---

**Require:** Actor-Critic Components
**Require:** Factor Encoder $\phi$
**Require:** Aggregation Module $\theta$
**Require:** Attention Module $\psi$
  1: Black Text: Standard Actor-Critic Algorithm
  2: Red Text: Proposed changes
  3: Cyan Text: Comments
  4: **for** each timestep $t = 1...T$ **do**
  5:     $z_t = \phi(s_t)$
  6:     $z_t^i = \phi(s_t)[i], \forall i \in \{1, \ldots, k\}$ *(b), (c)* in Figure 1
  7:     Compute state-value $V^\pi(s_t)$
  8:     Compute state values $V_i^\pi(z_t^i) \; \forall i \in \{1, \ldots, k\}$ *(d), (e)* in Figure 1
  9:     Compute state value $V^\pi(s_t) = \sum_{i=1}^k V_i^\pi(z_t^i)$ using Equation (9) *(f)* in Figure 1.
 10:     Compute $\alpha_t = \phi(z_t^1, \cdots, z_t^k, V^\pi(z_t^1), \cdots, V^\pi(z_t^k))$ using Equation (3) *(h), (i)* in Figure 1.
 11:     Compute $z_t = \theta(z_t^1, \cdots, z_t^k, \alpha_t^1, \cdots, \alpha_t^k)$ *(k)* in Figure 1.
 12:     Sample $a_t \sim \pi(z_t)$
 13:     Sample reward $r_t$ and next state $s_{t+1}$
 14:     Compute TD error $\delta_t = r_t + \gamma \times V^\pi(s_{t+1}) - V^\pi(s_t)$ using Equation (9).
 15:     Update value function and the Factor Encoder using the critic loss (TD error)
 16:     Update policy, *Attention Module* and *Aggregation Module* a using the actor loss (Policy-Gradient Theorem)
 17: **end for**

---

factored representations using an *Aggregation Module*, denoted by $\theta$. Essentially, the *Aggregation Module* returns a distribution of *weights* (denoted as $\{w_t^1 \cdots w_t^k\}$) over the factored representation, which are used to compute a weight sum of the factors.

In practice, we normalize the weights using the softmax operator, to ensure that they sum up to 1. The resulting weights, $\{a_t^1 \cdots a_t^k\}$, can be interpreted as some form of *attention* weights as they *control* how much the policy *attends* to any given factor. The aggregated representation (that will be fed to the policy network) is computed as $\theta(z_t^1, \cdots, z_t^k, a_t^1, \cdots, a_t^k,) = \sum_{i=1}^k z_t^i \times \alpha_t^i$ where $\times$ denotes the scalar product. This factor aggregation operation is shown as the component $k$ in Figure 1.

So far, we have described the high-level mechanism to aggregate the factored representations before passing to the policy network. In the next section (Section 4.1), we describe how we can obtain the attention scores for aggregating the representations.

## 4.1 Attending to the Factorized Representations

Given a collection of factor representations $[z_t^1, \cdots, z_t^k]$, we want to learn attention scores $[\alpha_t^1, \cdots, \alpha_t^k]$ (represented jointly as $\alpha_t$) that capture the relative importance of the different factors for the given task. These attention scores would enable the policy to determine the most relevant factors and attend to them more strongly than the less relevant ones.

There are several ways of computing $\alpha_t$. In the simplest case, we can set $\alpha_t^i = \frac{1}{k}$, i.e. *assign* equal weight to all the factors. We note that this simple instantiation of the proposed AFaR approach is competitive with the baseline approaches for simpler environments. However, it does not perform well on more complex environments. This limitation likely stems from the fact that not all factors are relevant at all times, making the *equal-weight* strategy sub-optimal. For example, if the agent is searching for a key to open a door, it may not need to attend to the door till it finds the key.

A more systematic and expressive approach is to learn the attention scores using an *Attention Module*, denoted using $\psi$ that takes as input the factor representations, $[z_t^1, \cdots, z_t^k]$ [4], and outputs the attention scores

---

[4]In practice, we concatenate the different factored representations. Other choices, like averaging over the factored representation, before feeding to the *Attention Module* should also work.

$[\alpha_t^1, \cdots, \alpha_t^k]$. The learnt attention scores enable the *Aggregation Module* to condition only on the important factors, while *ignoring* the irrelevant factors.

The *Attention Module* can be instantiated as a feedforward network that is trained using the actor loss, along with the policy network. Now we can compute the attention scores as:

$$\alpha_t = \psi(z_t^1, \ \cdots, z_t^k), \tag{2}$$

In the next section (Section 4.2), we describe how we can leverage the structure of Reward Factored MDP when aggregating the factored representations.

### 4.2 Leveraging the structure of Factored Reward MDP

Actor-critic algorithms commonly train the state value function, $V$, as one of the many components. The output of this state value function captures the *value* of a given state, as measured in terms of the expected returns. Comparing the state value functions for two states can determine which state will likely lead to higher returns for the learning agent. Extending this argument to the state factors, a factor with a higher value of the state value function will lead to a higher expected return for the agent than a factor with a lower value. This observation motivates the use of the state value of the factors as an input to the *Attention Module*. We compute the attention scores as:

$$\alpha_t = \psi(z_t^1, \ \cdots, z_t^k, V(z_t^1), \ \cdots, V(z_t^k)). \tag{3}$$

In practice, we concatenate the different factored representations, and the output of the state value function, corresponding to the different factors before feeding to the *Attention Module* should also work. The factored representations and the state values are *detached* from the computation graph before passing to the *Attention Module*. This ensures that only the critic loss updates the value function and the (factored) representations. Effectively, the critic controls what the different factors learn (via the critic loss), and the actor controls which factors are used for computing an action (via the policy loss). [5]

## 5 Experiments

So far, we have described how to learn factor representations and use them when training RL algorithms. In this section, we empirically verify the usefulness of the AFaR technique by designing experiments to answer the following questions: (i) Will the RL policy learned using the factored representation be useful? (ii) Does AFaR improve generalization to new environments where factors vary in novel ways? Here, "new environments" refer to environments with different layouts (or sizes) or of a different level of difficulty than the original environments. "Factors vary in novel way" refer to change in properties of the factors that the agent saw during training. For example, in the BigFish environment (from ProcGen), the number/size/shape of fishes varies across the environments. Similarly, in the MiniGrid environment, the number of doors and their types (i.e. locked or unlocked) varies across environments. (iii) How robust is AFaR to the number of factors that the agent learns?

We use the Procgen benchmark (Cobbe et al., 2020) and the MiniGrid environments (Chevalier-Boisvert et al., 2018) to evaluate the effectiveness of the proposed AFaR algorithm. Both environments are designed to assess RL algorithms for generalization and sample efficiency. Additionally, they satisfy critical criteria like high diversity across the environment instances, optimized implementation for fast training, and shared observation and action spaces. These environments are further described in Section 5.3 and Section 5.4.

---

[5]Following the same line of reasoning, the factored representations are detached from the computational graph even in the case of Equation (2).

### 5.1 Choice of Actor-Critic Methods

In Section 4, we noted that AFaR is a representation learning algorithm, and we need to pair it with a policy optimization algorithm for end-to-end reinforcement learning. In Algorithm 1, we highlighted the changes (in red color) for incorporating the AFaR with standard actor-critic algorithms. In this section, we describe the different actor-critic algorithms that we use in conjunction with AFaR and outline the rationale for choosing these specific baselines.

Our first criterion for selecting the baseline actor-critic algorithms was a strong performance on the selected environments. For the Procgen environments, we use two baselines: (i) Data-regularized Actor-Critic (DrAC) Raileanu et al. (2020), and (ii) Invariant Decoupled Advantage Actor-Critic (IDAAC) Raileanu & Fergus (2021). The key difference between these two approaches is that IDAAC is the state-of-the-art among methods that use a separate policy and value function, while DrAC is the state-of-the-art among methods that use a shared network for policy and value functions. For MiniGrid environments, we use Rewarding Impact-Driven Exploration (RIDE) Raileanu & Rocktäschel (2020) method, a state-of-the-art method for tasks on the MiniGrid environments.

There are additional benefits when using these methods as the underlying actor-critic methods for the AFaR approach. These methods incorporate different inductive biases and focus on different challenges in training RL agents. For example, DrAC uses data-augmentation techniques, IDAAC uses disjoint policy and value functions, and RIDE is designed to address exploration-related challenges. Moreover, these methods also differ in terms of how the value functions and the encoders are trained. For example, IDAAC learns a value function to provide the learning signal for an advantage function, which, in turn, is used for training the policy. RIDE uses two sets of encoders – one for training the policy and the other for computing the intrinsic rewards (used for computing the exploration bonuses). Integrating AFaR with such diverse methods and demonstrating that integrating with AFaR can improve over these baselines shows that our proposed algorithm is helpful for a variety of actor-critic algorithms.

For additional details about the underlying actor-critic methods, refer to Appendix E.

### 5.2 Integrating with Actor-Critic Methods

Integrating AFaR with existing actor-critic methods requires replacing the given encoder (from the underlying actor-critic method) with a *Factor Encoder*. A simple approach for instantiating a *Factor Encoder* is to start with the given encoder and scale the output dimension by a factor of $k$, where $k$ is the number of factors that we want to learn. Since the factored representations are aggregated using a weighted-sum, the size of the subsequent (and even the previous) layers does not have to be changed. Using AFaR increases the number of trainable parameters in the network. For the RIDE baseline, the number of parameters increase from $18M$ to $18M + 1.05 \times (k-1)M$, i.e. an increase of about 5.8% parameters for each new factor. For the DrAC baseline, the increase is from $626K$ to $626K + 66K \times (k-1)$, i.e. an increase of about 10.5% parameters for each new factor. For the IDAAC baseline, the increase is from $1.25M$ to $1.25M + 66K \times (k-1)$, i.e. an increase of about 5.2% parameters for each new factor.

When training the combined models, we take care to not change any hyperparameters (like learning rate, weight decay, exploration bonuses etc) for the underlying actor-critic algorithms. This choice ensures a fair-comparison as the combined model is not likely to perform better because of use of "better" hyperparameters. Any improvement in the performance has to come because of use of AFaR approach. This is also useful for demonstrating that AFaR can be used with the existing baselines without having to retune all the hyperparameters from scratch, which could have raised the bar of adoption for the proposed approach. We run all experiments with 10 seeds and report the mean and the standard error for all the results.

### 5.3 Progen Benchmark

#### 5.3.1 Environment

The Procgen benchmark comprises 16 procedurally generated environments as shown in Figure 2. Each environment represents

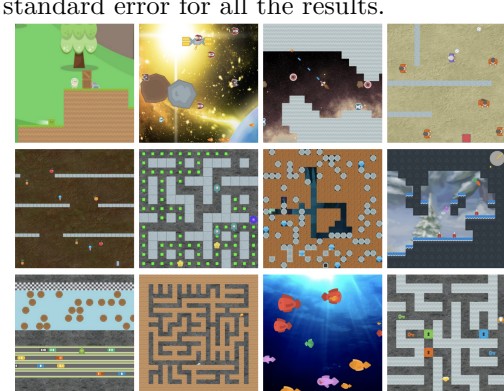

a distribution of *levels* where each *level* corresponds to a new environment, allowing for a strict partition between train and test environments. Each environment has multiple objects and entities that the agent can interact with. For example, in the BigFish environment, the agent comes across many fishes (of different shapes, sizes and color). In this case, the AFaR agent can learn factors (representations) corresponding to the different fishes.

Following the setup in Raileanu et al. (2020), we train the agent on a fixed set of 200 levels while testing on the full distribution of levels. In practice, this is simulated by sampling levels at random during evaluation. For additional details about the environment, refer Appendix B.1.

### 5.3.2 Baselines

DrAC builds on the idea of using data augmentation for improving generalization in RL (Laskin et al., 2020; Srinivas et al., 2020; Kostrikov et al., 2020), and introduces regularization terms that make the use of data augmentation theoretically sound for actor-critic algorithms. We replace the shared observation encoder with our *Factored Encoder*. IDAAC proposes using separate networks to model the policy and the value functions by suggesting that using a shared representation (for actor and critic) can lead to overfitting. The learnt value function is used for training an advantage function which acts as the critic for the policy. Further, the advantage function shares parameters with the actor network, as is the case for standard actor-critic architectures. So we factorize the observation encoder of the policy (that is shared with the advantage function). For the IDAAC architecture, "critic" (as used in Algorithm 1 and Figure 1) refers to the advantage function. For additional details on the baselines, refer to Appendix E.

### 5.3.3 Results

We train individual policies on each of the 16 environments using 200 levels (per environment) and report the score on the evaluation environments, after training for $25M$ (environment) steps. The agent is evaluated on the full distribution of the levels (generated by sampling levels at random during evaluation). Along side reporting the mean and the standard error metrics (computed over 10 seeds), we also report environments where the improvements are statistically significant. Details about testing for statistical significance can be found in Appendix D.

**DrAC** In Table 1, we compare the performance of DrAC (as a baseline) with AFaR (using DrAC for learning the policy). We observe that the proposed AFaR approach leads to statistically significant improvement in 7 out of 16 environments. The corresponding training and evaluation plots are presented in Figure 6 and Figure 7 (in Appendix). We also perform an ablation where we train the systems using just 10 levels (instead of 200 levels) which severely

| Environment | DrAC | AFaR |
|---|---|---|
| Bigfish | $8.88 \pm 0.89$ | $\mathbf{12.53 \pm 0.53}$ |
| Bossfight | $7.77 \pm 0.21$ | $7.79 \pm 0.21$ |
| Caveflyer | $4.37 \pm 0.21$ | $\mathbf{5.63 \pm 0.25}$ |
| Chaser | $6.56 \pm 0.24$ | $7.07 \pm 0.22$ |
| Climber | $6.76 \pm 0.14$ | $\mathbf{7.14 \pm 0.1}$ |
| Coinrun | $8.62 \pm 0.06$ | $8.62 \pm 0.09$ |
| Dodgeball | $4.78 \pm 0.23$ | $5.24 \pm 0.18$ |
| Fruitbot | $27.85 \pm 0.28$ | $28.22 \pm 0.15$ |
| Heist | $3.96 \pm 0.12$ | $\mathbf{4.93 \pm 0.15}$ |
| Jumper | $5.8 \pm 0.11$ | $5.58 \pm 0.09$ |
| Leaper | $4.0 \pm 0.3$ | $\mathbf{5.19 \pm 0.33}$ |
| Maze | $6.33 \pm 0.11$ | $\mathbf{6.72 \pm 0.07}$ |
| Miner | $9.63 \pm 0.13$ | $\mathbf{10.11 \pm 0.09}$ |
| Ninja | $5.41 \pm 0.13$ | $5.49 \pm 0.11$ |
| Plunder | $8.12 \pm 0.48$ | $9.27 \pm 0.34$ |
| Starpilot | $29.67 \pm 0.85$ | $28.89 \pm 0.65$ |

Table 1: We compare AFaR with DrAC (as the baseline approach) on the evaluation environments after training for $25M$ steps. We report the mean and the standard error over 10 runs. The results marked in **bold** denote the results that are statistically significantly better. Additional details about testing for statistical significance can be found in Appendix D.

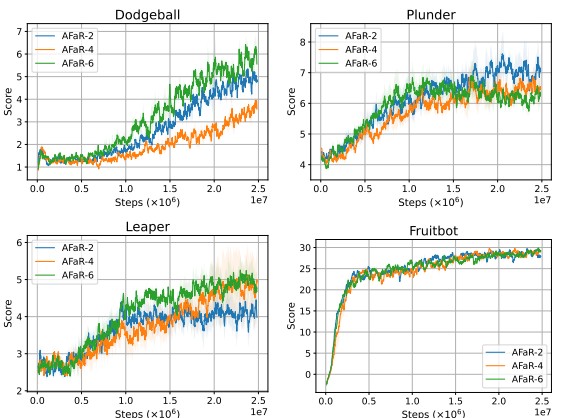

limits the diversity in the training data and can hurt the generalization performance significantly. In Figure 8 (in Appendix), we show that while the generalization performance degrades for both the approaches, AFaR still outperforms (or is at par) the baseline approach on all the environments.

We compare the performance of AFaR with DrAC (as the baseline approach) on the evaluation environments from Procgen Benchmark, when training with 10 levels (per environment) for $25M$ steps. The solid curves correspond to the mean performance (score) while the shaded regions represent the standard error (computed over 10 runs). For corresponding curves when training on 200 levels per environment, refer to Figure 7.

AFaR introduces a new hyper-parameter $k$, or the number of representations that the *Factor Encoder* produces. We perform an ablation experiment to understand how sensitive is the performance of AFaR algorithm to this hyperparameter. In Figure 3, we note that in some cases, the choice of $k$ does not make much difference in the convergence performance for the agent. However, in general, the performance can be improved by tweaking $k$.

**IDAAC** In Table 2, we compare the performance of IDAAC (as a baseline) with AFaR (using IDAAC for learning the policy) using the same experiment protocol as before. Recall that IDAAC is an actor-critic method that uses separate policy and value functions. We observe that the proposed AFaR approach improves over the baseline in 14 out of 16 environments (statistically significant improvement in 4 environments). We also consider two additional baselines. In the first case, we factorize only the actor and not the critic. The actor is factorized as before but the critic produces a single state-value, using the concatenated factors as the feature. The agent assigns weight to the factors by using another neural network. This agent is refered to as the Factorized Actor. In the second case, we *disable* the *Attention Module* and assign equal weights to all the factors while aggregating them. We denote this ablation as AFaR-mean. As we observe in Table 2, Factorized Actor sometimes improves over the IDAAC baseline but generally lags behind IDAAC. AFaR-mean is generally stronger than Factorized Actor but lags behind AFaR.

### 5.4 MiniGrid

#### 5.4.1 Environment

MiniGrid is a procedurally generated grid-world environment where the "world" is a $N \times M$ grid of tiles. Each tile may contain atmost one object and the learning agent has to navigate the grid and find and interact with objects to complete different tasks. In these environments, the AFaR agent can learn factors (representations) corresponding to the different objects. We use the following environments from MiniGrid:

1. `MiniGrid-KeyCorridorSxRy` environments where `x` denotes the size of a room and `y` denotes the number of rows — for compactness we will denote these by `KeyCorridorSxRy`. The agent's task is to pick up an object behind a locked door. The key to this locked door is hidden in another room, and the agent has to explore the environment to find the hidden key.

2. `MiniGrid-MultiRoomNxSy` environments where `x` denotes the number of room and `y` denotes the maximum size of a room — for compactness we will denote these by `MultiRoomNxSy`. The agent is placed in the first room and must navigate to a green goal square in the most distant room from the agent

For additional details about the environment, refer Appendix B.2.

| Environment | IDAAC | Factorized Actor | AFaR-mean | AFaR |
|---|---|---|---|---|
| Bigfish | $17.2 \pm 0.8$ | $16.99 \pm 1.2$ | $18.4 \pm 0.6$ | $\mathbf{20.9 \pm 1.5}$ |
| Bossfight | $9.8 \pm 0.5$ | $10.23 \pm 0.5$ | $10.6 \pm 0.4$ | $\mathbf{10.77 \pm 0.8}$ |
| Caveflyer | $5.25 \pm 0.5$ | $5.37 \pm 0.8$ | $5.25 \pm 0.7$ | $\mathbf{5.5 \pm 0.5}$ |
| Chaser | $7.4 \pm 1.0$ | $7.45 \pm 0.8$ | $7.2 \pm 1.4$ | $\mathbf{8.1 \pm 0.9}$ |
| Climber | $9.3 \pm 0.4$ | $8.32 \pm 0.6$ | $8.5 \pm 0.5$ | $9.4 \pm 0.3$ |
| Coinrun | $9.5 \pm 0.4$ | $9.5 \pm 0.27$ | $9.25 \pm 0.4$ | $9.75 \pm 0.2$ |
| Dodgeball | $3.1 \pm 0.3$ | $3.32 \pm 0.5$ | $3.15 \pm 0.8$ | $\mathbf{4.3 \pm 0.5}$ |
| Fruitbot | $28.8 \pm 0.9$ | $28.47 \pm 0.92$ | $29.1 \pm 1.5$ | $\mathbf{30.3 \pm 1.1}$ |
| Heist | $3.5 \pm 0.4$ | $3.1 \pm 0.5$ | $3 \pm 0.9$ | $\mathbf{3.75 \pm 0.5}$ |
| Jumper | $7.0 \pm 0.8$ | $6.25 \pm 0.46$ | $6.75 \pm 0.41$ | $7.2 \pm 0.7$ |
| Leaper | $7.75 \pm 1.1$ | $7.75 \pm 0.7$ | $8 \pm 0.9$ | $\mathbf{8.5 \pm 0.6}$ |
| Maze | $6.3 \pm 0.3$ | $5.25 \pm 0.6$ | $5 \pm 0.8$ | $\mathbf{6.7 \pm 0.3}$ |
| Miner | $10.34 \pm 0.6$ | $6.22 \pm 1.3$ | $8.94 \pm 0.8$ | $10.16 \pm 0.5$ |
| Ninja | $7.5 \pm 0.9$ | $7.37 \pm 0.7$ | $7.75 \pm 0.5$ | $\mathbf{8.25 \pm 0.6}$ |
| Plunder | $23.9 \pm 1.1$ | $.\ 22.71 \pm 1.8$ | $21.7 \pm 1.7$ | $\mathbf{24.39 \pm 1.2}$ |
| Starpilot | $41.65 \pm 2.1$ | $37.91 \pm 1.9$ | $38.82 \pm 1.8$ | $\mathbf{42.35 \pm 2.2}$ |

Table 2: We compare AFaR with IDAAC on the evaluation environments after training for $25M$ steps. We report the mean and the standard error over 10 runs. The results marked in **bold** denote the results that are (i) best performing among all the approaches and (ii) are statistically significantly better than IDAAC or AFaR or both (whichever applicable). Additional details about testing for statistical significance can be found in Appendix D.

### 5.4.2 Baselines

MiniGrid environments use a sparse reward setup which makes exploration a key challenge. RIDE proposed a novel intrinsic reward mechanism in which the agent is rewarded for actions that affect its learned state representation. On the MiniGrid environments, RIDE improves sample efficiency compared to approaches like count-based exploration Bellemare et al. (2016), Random Network Distillation Burda et al. (2019), Intrinsic Curiosity Module Pathak et al. (2017) and IMPALA (Espeholt et al., 2018).

We note that RIDE proposed using two sets of encoders — one for the actor-critic method and the other for computing the intrinsic reward. The first encoder is trained along the lines of the standard RL works, while the second encoder is trained using forward and backward dynamics loss. Since the *Factor Encoder*, $\phi$, is proposed in the context of actor-critic algorithms, we only replace the first encoder with the *Factor Encoder* and do not change the design of the second encoder (that is used for the computation of the intrinsic reward). This design choice also enables a fairer comparison between RIDE and the proposed approach as we do not change how the intrinsic rewards are computed (which is a core contribution of the RIDE approach). Further, we modify the actor-critic components of RIDE as per Algorithm 1. RIDE uses a recurrent network for maintaining a history over the observations. We share the weights of the recurrent network across the different factors (with every factor having their own hidden state).

### 5.4.3 Results

With the MiniGrid environments, we look at two forms of generalization. In the first case, we train and evaluate the agents on a given environment. Note that every time the environment is reset, a new instance of the environment is sampled, with a new randomized layout and new arrangement of objects. The agent also spawns at a random location in the grid. This ensures that the learning agent can not solve the task during evaluation by simply *memorizing* the training instances, as not only all the training instances unique (which makes memorization very challenging), the test instances are disjoint from the training instances. We perform this evaluation with the `KeyCorridorS3R3-v0` environment and the `MultiroomN7S4-v0` environment. In both the cases, we report the number of environment interactions (in millions) that a learning agent needs to converge. For `KeyCorridorS3R3-v0` environment, the success rate at convergence is 90% while for `MultiroomN7S4-v0` environment, the convergence performance is 75%. Note that for both the environments, the convergence performance of the agents is the same though the sample complexity is different. In Table 3,

we note that the AFaR approach is about 18% more sample efficient that RIDE for `KeyCorridorS3R3-v0` and about 33% more sample efficient for the `MultiroomN7S4-v0` environment (In Table 4).

Similar to the IDAAC setup, we consider two additional baselines. In the first case, we factorize only the actor and not the critic. The actor is factorized as before but the critic produces a single state-value, using the concatenated factors as the feature. The agent assigns weight to the factors by using another neural network. This agent is referred to as the Factorized Actor. In the second case, we *disable* the *Attention Module* and assign equal weights to all the factors while aggregating them. We denote this ablation as AFaR-mean. We note that while AFaR-mean provides a marginal improvement over RIDE, it lags behind AFaR. Factorized Actor needs significantly more steps to solve the tasks. The corresponding evaluation curves can be found in Figure 9.

Similar to the IDAAC setup, we consider two additional baselines. In the first case, we factorize only the actor and not the critic. The actor is factorized as before but the critic produces a single state-value, using the concatenated factors as the feature. The agent assigns weight to the factors by using another neural network. This agent is referred to as the Factorized Actor. In the second case, we *disable* the *Attention Module* and assign equal weights to all the factors while aggregating them. We denote this ablation as AFaR-mean. As we observe in Table 2, AFaR outperforms all the baselines. AFaR-mean is sometimes (slightly) better tha RIDE while under-performing on one environment. Factorized Actor lags behind all the other methods.

In the second case, we train the agent on one environment and evaluate it on a different environment in a zero-shot manner. Specifically, we perform zero-shot evaluation, at regular intervals, on `KeyCorridorS3R1-v0` and `KeyCorridorS3R2-v0` environments, while training on the `KeyCorridorS3R3-v0` environment and on `MultiroomN10S4-v0` environment while training on the `MultiroomN7S4-v0` environment. Following the protocol in the previous set of experiments, we report the number of environment interactions (in millions) that a learning agent needs to converge. For `KeyCorridorS3R1-v0` and `KeyCorridorS3R2-v0` environments, the zero-shot success rate at convergence is 90% while for `MultiroomN10S4-v0` environment, the zero-shot the convergence performance is 75%. This second setup is more challenging than the

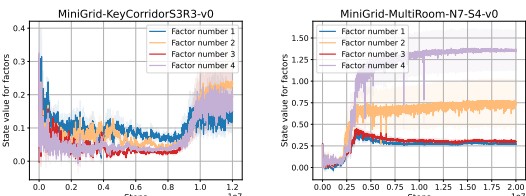

Figure 4: We plot how the state-value for the different factors, averaged across steps in an episode and across a batch of environments, changes over time for the `KeyCorridorS3R3-v0` environment and the `MultiroomN7S4-v0` environments (left to right).

first setup as now even the size of the rooms and/or the grid changes, along with the layout. The corresponding results are shown in the second and third rows of Table 3 and in the second row of Table 4 where we record the number of environment interactions (in millions) on `KeyCorridorS3R3-v0` and `MultiroomN7S4-v0` respectively. We note that the AFaR approach requires much fewer samples to generalize to the unseen environments. The corresponding evaluation curves can be found in Figure 9.

We want to ensure that: (i) the factors are not degenerate, i.e., most or all-but-one factors are not collapsing to 0 value, and (ii) AFaR is not learning to simply average the state-value of the factors to compute the overall state value. In Figure 4, we plot the state-value for the different factors, averaged across steps in an episode and across a batch of environments, changes over time for the `KeyCorridorS3R3-v0` environment and the `MultiroomN7S4-v0` environments (left to right). As we can see in the figure, all the factors are assigned a non-zero (state) value (i.e., the factors do not collapse to 0 value), and all the factors are not contributing uniformly to the value function (i.e., the overall state value is not simply a mean of the value corresponding to the different factors). Moreover, the distribution of the state value (across factors) seems to depend on the task. For the `KeyCorridorS3R3-v0` task, the agent has to interact with multiple values, and the value assigned to the different factors is quite close. The `MultiroomN7S4-v0` task does not focus on interacting with multiple objects, so the distribution (of state values) is more uneven.

In Figure 4, we plot how the state-value for the different factors, averaged across steps in an episode and across a batch of environments, changes over time for the `KeyCorridorS3R3-v0` environment and the `MultiroomN7S4-v0` environments (left to right).

| Environment | AFaR | RIDE | AFaR-mean | Factorized Actor |
|---|---|---|---|---|
| | Number of environment interations (in millions) | | | |
| MiniGrid-KeyCorridorS3R3-v0 | 8.5 | 10.4 | 10.3 | 13.6 |
| MiniGrid-KeyCorridorS3R1-v0 | 7.6 | 8.9 | 8.9 | 9.05 |
| MiniGrid-KeyCorridorS3R2-v0 | 8.5 | 9.2 | 10.1 | 10.2 |

Table 3: We compare the performance of AFaR with RIDE (baseline approach) on `KeyCorridorS3R3-v0`, `KeyCorridorS3R1-v0` and `KeyCorridorS3R2-v0` environments (top to bottom) in terms of the number of environment interactions (in millions) that the agent needs to converge to the 90% success rate. We also include an ablation version of the AFaR algorithm where we *disable* the *Attention Module* and assign equal weights to all the factors while aggregating them. We denote this ablation as AFaR-mean. Note that while the agent is evaluated on three environments (`KeyCorridorS3R3-v0`, `KeyCorridorS3R2-v0`, `KeyCorridorS3R1-v0`), it was trained only on the `KeyCorridorS3R3-v0` environment, thus the evaluation on `KeyCorridorS3R2-v0` and `KeyCorridorS3R1-v0` environments is done in a zero-shot manner. For all the three environments, AFaR obtains the best sample efficiency. The corresponding evaluation curves can be found in Figure 9.

| Environment | AFaR | RIDE |
|---|---|---|
| | Number of environment interations (in millions) | |
| MiniGrid-MultiroomN7S4-v0 | 6.8 | 9.03 |
| MiniGrid-MultiroomN10S4-v0 | 7.6 | 9.7 |

Table 4: We compare the performance of AFaR with RIDE (baseline approach) on `MultiroomN7S4-v0` and `MultiroomN10S4-v0` environments (top to bottom) in terms of the number of environment interactions (in millions) that the agent needs to converge to the 75% success rate. Note that while the agent is evaluated on two environments (`MultiroomN7S4-v0`, `MultiroomN10S4-v0`, it was trained only on the `MultiroomN7S4-v0` environment, thus the evaluation on `MultiroomN10S4-v0` environment is done in a zero-shot manner. For both the environments, AFaR obtains the best sample efficiency.

## 6 Related Work

Factored Reward MDPs are closely related to Factored MDPs and Successor Features, which have a rich history of prior work. Our proposed approach is also related to the prior work on linear approximation of value functions and mixture of experts. Given the focus of our approach is on improving generalization in RL, we briefly summarize the related efforts in the literature.

**Factored MDPs** (Boutilier et al., 1995; 1999; Kearns & Koller, 1999; Cui & Khardon, 2016) are a special class of MDPs where the state, dynamics, and the reward can be factored into variables. Kearns & Koller (1999) proposed modeling the factorized transition function as a Dynamic Bayesian Network (DBN) where the underlying structure is known upfront, but the parameters are to be inferred. Guestrin et al. (2003) proposed two algorithms, based on linear programming and policy iteration, for learning in Factored MDPs. More recently, Factored MDPs have been used in conjunction with deep RL approaches in environments known to have factored structure and shown to improve sample efficiency and transfer (Li & Czarnecki, 2018; Wang et al., 2018; Balaji et al., 2020). However, factored transitions are a strong assumption and do not necessarily give rise to factored value functions. Our work is also related to the work on Exogenous MDPs (Dietterich et al., 2018) which proposes factorizing an MDP into exogenous Markov Reward Process and an endogenous Markov Decision Process. It further shows that if the reward function is assumed to decompose additively into the two factors, the Bellman equation for the original MDP decomposes into two equations: one for the exogenous Markov reward process (Exo-MRP) and the other for the endogenous MDP (EndoMDP). AFaR does not restrict the decomposition to exogenous and endogenous factors and supports using a much larger number of factors.

We operate on Factored Reward MDPs, which, as shown in Section 2.2, admit factorized value functions. Such factorization is related to the work on **Successor Features** Dayan (1993); Barreto et al. (2017). There, the key idea is to represent the value functions as the linear combination of a special class of basis functions. These basis functions, referred to as the successor features, encode the state such that under a policy, the representation of the state is similar to the representation of the successor states. However, Successor Features are commonly used to obtain fast policy evaluation to aid transfer across different reward functions rather than to improve generalization in the single-task settings Lehnert et al. (2017); Barreto et al. (2018). Other works that represent the **value functions as a linear combination of basis functions** include Parr et al. (2008) that assumes that the transition function is known and uses linear fixed-point methods, Mahadevan & Maggioni (2006), Lagoudakis & Parr (2003) and Konidaris et al. (2011) that used Polynomial basis, proto-value functions and Fourier basis respectively. Koller & Parr (1999) is quite close to our work as they also proposed imposing the additive structure on the value function. However, they use hand-designed basis functions while we learn factorization as part of the training process. Approximating the value function (of a state) as a sum of the value function of its factors can be seen as an application of the *mean-field principle* (Anderson & Peterson, 1987; Peterson, 1987; Parisi, 1988; Jordan et al., 1999; Wainwright & Jordan, 2008; Peyrard & Sabbadin, 2006). Mean field principle has also been used in context of multi-agent RL (Yang et al., 2018; Subramanian et al., 2020), where the interactions within the population of agents are approximated as those between a single agent and the average effect from the population of neighboring agents.

Our work is also related to the work on **Mixture of Experts** (MoE) (Jacobs et al., 1991; Jordan & Jacobs, 1994; Chen et al., 1999; Yuksel et al., 2012) where the high level idea is to learn *experts* that specialize to different aspects of a task/data distribution. Aljundi et al. (2017); Rannen et al. (2017) proposed learning per-task experts for avoiding catastrophic forgetting in the lifelong learning setup (McCloskey & Cohen, 1989; Silver et al., 2013; Mitchell et al., 2018; Sodhani et al., 2022a). MoE architectures are also prominently used in natural language processing (Shazeer et al., 2017; Lepikhin et al., 2020; Fedus et al., 2021; Lewis et al., 2021) and computer vision (Ahmed et al., 2016; Gross et al., 2017; Yang et al., 2019; Wang et al., 2020). In the RL literature, MoE has been used in the context of multi-task learning (Yang et al., 2020; Sodhani et al., 2021), hierarchical reinforcement learning (Andreas et al., 2017; Goyal et al., 2019), multi-modal policy (Ren et al., 2021) and multi-agent learning (He et al., 2016). In contrast to these works, we use a MoE-like architecture to map a given MDP to an approximate Factored Reward MDP and train RL agents on this approximate Factored Reward MDP.

**Generalization.** Given that Factored Reward MDPs are more *compact* than standard MDPs and exhibit factored value functions, we hypothesize that learning a Factored Reward MDP will improve the generalization performance of downstream RL algorithms. Previous works have focused on different aspects of generalization: Bellemare et al. (2013); Tassa et al. (2018) focused on the singleton environment setup where the train and the test environments are the same while Cobbe et al. (2020); Küttler et al. (2020); Samvelyan et al. (2021); Zhang et al. (2021a) focused on the setup where the train and the evaluation distributions are the same, even though the instances of train and evaluation environments are different. Ahmed et al. (2020); Stone et al. (2021) focused on the setup where the train and evaluation distributions differ. Along a different axis, Sodhani et al. (2022b) focused on the zero-shot generalization setup while Rakelly et al. (2019) allow few gradient updates (or limited interaction with the environment) before evaluating the generalization performance. In this work, we consider both setups — where the train and the evaluation distribution may or may not be the same. We focus on the zero-shot generalization setup as it is a more challenging and realistic setup. For example, an RL agent deployed in the real world may not know when the underlying task distribution has changed and may have to generalize to the new distribution in a zero-shot manner.

## 7 Conclusion

In this work, we propose to study a form of structured environment with additively factorized rewards. We call this setup *Factored Reward MDP*. A nice feature of this structured MDP is that the value function also factorizes. We design an algorithm, AFaR, that learns to map a given MDP to an approximate Factored Reward MDP. We show that AFaR can be easily combined with existing RL algorithms, leading to improved sample efficiency and generalization performance in both Procgen benchmark and the MiniGrid environments. An interesting future work would be to extend AFaR to learn *factorized* policies that can act on the individual factors that may lead to benefits like better exploration.

Another interesting follow up direction would be to incorporate the factorized structure into a universal policy using hierarchical approaches. If the learned factored representation correspond to the *factors of variation* in an environment, the factors can be used as input to low-level policies, or options (Sutton et al., 1999) which can act on these factors independently thus aiding the interpretability of options and enabling improved exploration strategies.

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

# A   Q-Functions Under Factorized Rewards

We show that a Factored Reward MDP has state value functions and state-action value functions that can be written as a sum of factors.

**Proposition 2.2.** *Consider a Factored Reward MDP where the state $s_t$ is factorized into $k$ state factors $s_t^1, \cdots, s_t^k$ and the reward function is factorized as $R(s_t, a_t) = \sum_{i=1}^{k} r(s_t^i, a_t)$ where $r : \mathcal{S} \times \mathcal{A} \to \mathbb{R}$ are local reward functions. The state value function and the state-action value functions for any policy $\pi$ can be written as:*

$$V^\pi(s_t) = \sum_{i=1}^{k} V_i^\pi(s_t^i), \tag{4}$$

$$Q^\pi(s_t, a_t) = \sum_{i=1}^{k} Q_i^\pi(z_t^i, a_t), \tag{5}$$

*where*

$$V_i^\pi(s_t^i) = \sum_{t'=t}^{\infty} \gamma^{t'} \sum_{s_{t'+1}} T(s_{t'+1}|s_{t'}, a_{t'}) r(s_{t'}^i, a_{t'}),$$

$$Q_i^\pi(s_t^i, a_t) = r(s_t^i, a_t) + \sum_{t'=t+1}^{\infty} \gamma^{t'} \sum_{s_{t'+1}} T(s_{t'+1}|s_{t'}, a_{t'}) r(s_{t'}^i, a_{t'})$$

*can be seen as state value and the state-action value functions for the $i^{th}$ factor, respectively.*

*Proof.* The state-value function is defined as:

$$V^\pi(s_t) = \sum_{t'=t}^{\infty} \gamma^{t'} \sum_{s_{t'+1}} T(s_{t'+1}|s_{t'}, a_{t'}) R(s_{t'}, a_{t'}). \tag{6}$$

Applying the reward factorization, we get

$$V^\pi(s_t) = \sum_{t'=t}^{\infty} \gamma^{t'} \sum_{s_{t'+1}} T(s_{t'+1}|s_{t'}, a_{t'}) \sum_{i=1}^{k} r(s_{t'}^i, a_{t'}). \tag{7}$$

Re-arranging the terms, we get

$$V^\pi(s_t) = \sum_{i=1}^{k} \sum_{t'=t}^{\infty} \gamma^{t'} \sum_{s_{t'+1}} T(s_{t'+1}|s_{t'}, a_{t'}) r(s_{t'}^i, a_{t'}). \tag{8}$$

Equation 8 can be re-written as:

$$V^\pi(s_t) = \sum_{i=1}^{k} V_i^\pi(s_t^i), \tag{9}$$

where $V_i^\pi(s_t^i) = \sum_{t'=t}^{\infty} \gamma^{t'} \sum_{s_{t'+1}} T(s_{t'+1}|s_{t'}, a_{t'}) r(s_{t'}^i, a_{t'})$ can be seen as the state value function for the $i^{th}$ factor. Similarly, we can obtain the following factorization for the state-action value function:

$$Q^\pi(s_t, a_t) = \sum_{i=1}^{k} Q_i^\pi(z_t^i, a_t), \tag{10}$$

where

$$Q_i^\pi(s_t^i, a_t) = r(s_t^i, a_t) + \sum_{t'=t+1}^{\infty} \gamma^{t'} \sum_{s_{t'+1}} T(s_{t'+1}|s_{t'}, a_{t'}) r(s_{t'}^i, a_{t'})$$

and can be seen as the state-action value function for the $i^{th}$ factor. $\square$

## B   Environments

We use the Procgen benchmark (Cobbe et al., 2020) and MiniGrid environment (Chevalier-Boisvert et al., 2018) to evaluate the effectiveness of the proposed algorithm.

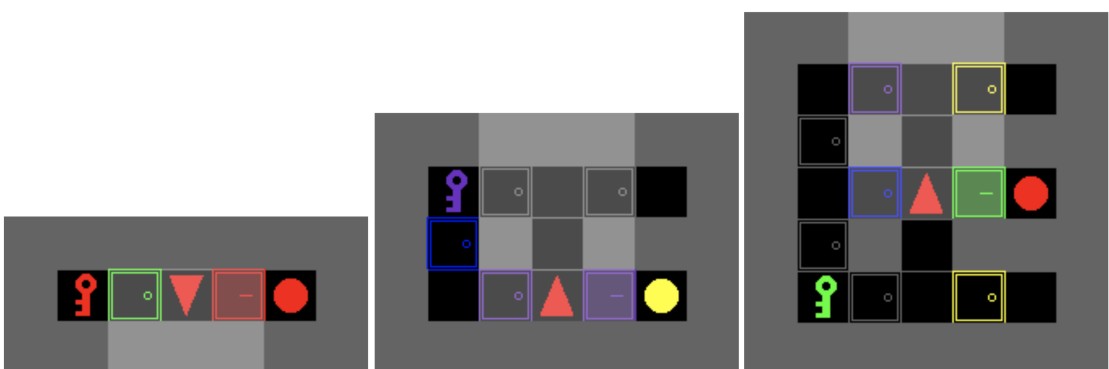

Figure 5: Screenshots for `KeyCorridorS3R1-v0`, `KeyCorridorS3R2-v0`, and `KeyCorridorS3R3-v0` environments (left to right). Taken from Chevalier-Boisvert et al. (2018).

## B.1 Procgen Benchmark

The Procgen benchmark comprises 16 procedurally generated environments designed to evaluate RL algorithms for generalization and sample efficiency. Each environment represents a distribution of *levels* where each *level* corresponds to a new environment, allowing for a strict partition between train and test environments. The benchmark satisfies several important criteria:

1. **High diversity** in the environments

2. **Optimized environments** that enable fast evaluation of algorithms

3. **Control over parameters** like difficulty and need for exploration. All environments support two modes - easy and hard.

4. **Emphasize on visual recognition and motor control**

5. **Level solvability** with about 99% levels solvable for each environment

6. **Shared observation and action space**

### B.1.1 Observation Space

The observation space is an RGB image of size $64 \times 64 \times 3$.

### B.1.2 Action Space

All environments use a discrete action space with 15 actions. In some environments, a *no-op* action is also present. All environments use a discrete 15-dimensional action space and produce $64 \times 64 \times 3$ RGB observations. We use Procgen's easy setup, so for each game, agents are trained on 200 levels and tested on the full distribution of levels. More details about our experimental setup and hyperparameters can be found in Appendix E.

## B.2 MiniGrid Environment

Minimalistic Gridworld Environment (MiniGrid) is an open source grid world Gym environment [6]. It enables the creation of gridworld environments with adjustable parameters (like size and layout of the grid) with different objects of different types, colors, and behaviors.

---

[6]https://github.com/maximecb/gym-minigrid

Table 5: Hyperparameter values that are common across all the experiments with Procgen Benchmark

| Hyperparameter | Hyperparameter values |
|---|---|
| optimizer | RMSProp |
| learning rate | $5 \times 10^{-4}$ |
| alpha for optimizer | 0.99 |
| epsilon for optimizer | $10^{-5}$ |
| discount | 0.999 |
| gae lambda parameter | 0.95 |
| entropy coefficient | 0.01 |
| value loss coefficient | 0.5 |
| max grad norm | 0.5 |
| number of training processes | 64 |
| number of forward steps in A2C | 256 |
| number of PPO epochs | 3 |
| number of minibatches for PPO | 8 |
| clip parameter for PPO | 0.2 |
| number of environment steps to train on | $25 \times 10^6$ |
| distribution mode for environment | easy |
| number of levels | 10, 100, 200, 300 |
| augmentation type | crop |
| coefficient for the augmentation loss | 0.1 |

Table 6: Hyperparameter values that are specific to the AFaR method, when trained on Procgen Benchmark

| Hyperparameter | Hyperparameter values |
|---|---|
| number of encoders | 2, 4, 6, 8 |

### B.2.1  Structure of the world

The "world" is a $N \times M$ grid of tiles. Each tile contains zero or one object. Each object has a color (discrete value) and a type (discrete value). Objects include - wall, door, key, box etc. The agent can perform interactions like "pick" or "carry" an object with some constraints. For example, the agent can carry at most one object at a time. Agent must carry a key (of matching color) to open a door.

### B.2.2  Observation Space

Each tile (in the grid world) is encoded as a 3 dimensional tuple of integers of the form $(object\_id, color\_id, state)$. The observations (for the agent) are egocentric and partial. i.e., the agent views the $3 \times 3$ sized grid in front of it. Moreover, the agent can not see across obstacles (like walls or closed doors).

### B.2.3  Action Space

The action space is discrete, with 7 actions: turn left, turn right, move forward, pick up an object, drop the object being carried, toggle (open doors, interact with objects) and done (task completed, optional).

Table 7: Best performing values for the number of encoders for the different environments and baselines, when trained on Procgen Benchmark

| Environment | AFaR with DrAC baseline | AFaR with IDAAC baseline |
|:---:|:---:|:---:|
| Bigfish | 4 | 4 |
| Bossfight | 6 | 6 |
| Caveflyer | 2 | 2 |
| Chaser | 4 | 4 |
| Climber | 4 | 4 |
| Coinrun | 4 | 4 |
| Dodgeball | 6 | 6 |
| Fruitbot | 6 | 6 |
| Jumper | 4 | 4 |
| Leaper | 6 | 6 |
| Maze | 4 | 4 |
| Miner | 6 | 6 |
| Ninja | 4 | 4 |
| Plunder | 2 | 2 |
| Starpilot | 6 | 6 |

### B.2.4 Reward Function

The agent is given a sparse reward for reaching the green goal tile. If the agent does not reach the tile within a specified time, it gets zero rewards. In case the agent completes the task, it gets a reward of $1 - 0.9 \times \frac{step\_count}{max\_steps}$. i.e., the agent is rewarded more when it completes the task faster.

Table 8: Hyperparameter values that are common across all the experiments with MiniGrid environments

| Hyperparameter | Hyperparameter values |
|:---|:---|
| total frames | 30000000 |
| entropy cost | 0.0005 |
| intrinsic reward coefficient | 0.1 |
| number of input frames | 1 |
| number of actors | 40 |
| batch size | 32 |
| unroll length | 100 |
| max grad norm | 40.0 |
| entropy cost | 0.001 |
| baseline cost | 0.5 |
| learning rate | 0.0001 |
| alpha (for optimizer) | 0.99 |
| momentum (for optimizer) | 0.0 |
| epsilon (for optimizer) | $10^{-5}$ |
| forward loss coefficient (for exploration) | 10.0 |
| inverse loss coefficient (for exploration) | 0.1 |
| intrinsic reward coefficient (for exploration) | 0.5 |
| random loss coefficient (for exploration) | 0.1 |

Table 9: Hyperparameter values that are specific to the AFaR method, when trained on MiniGrid environments

| Hyperparameter | Hyperparameter values |
|---|---|
| number of encoders | 2, 4, 6, 8 |

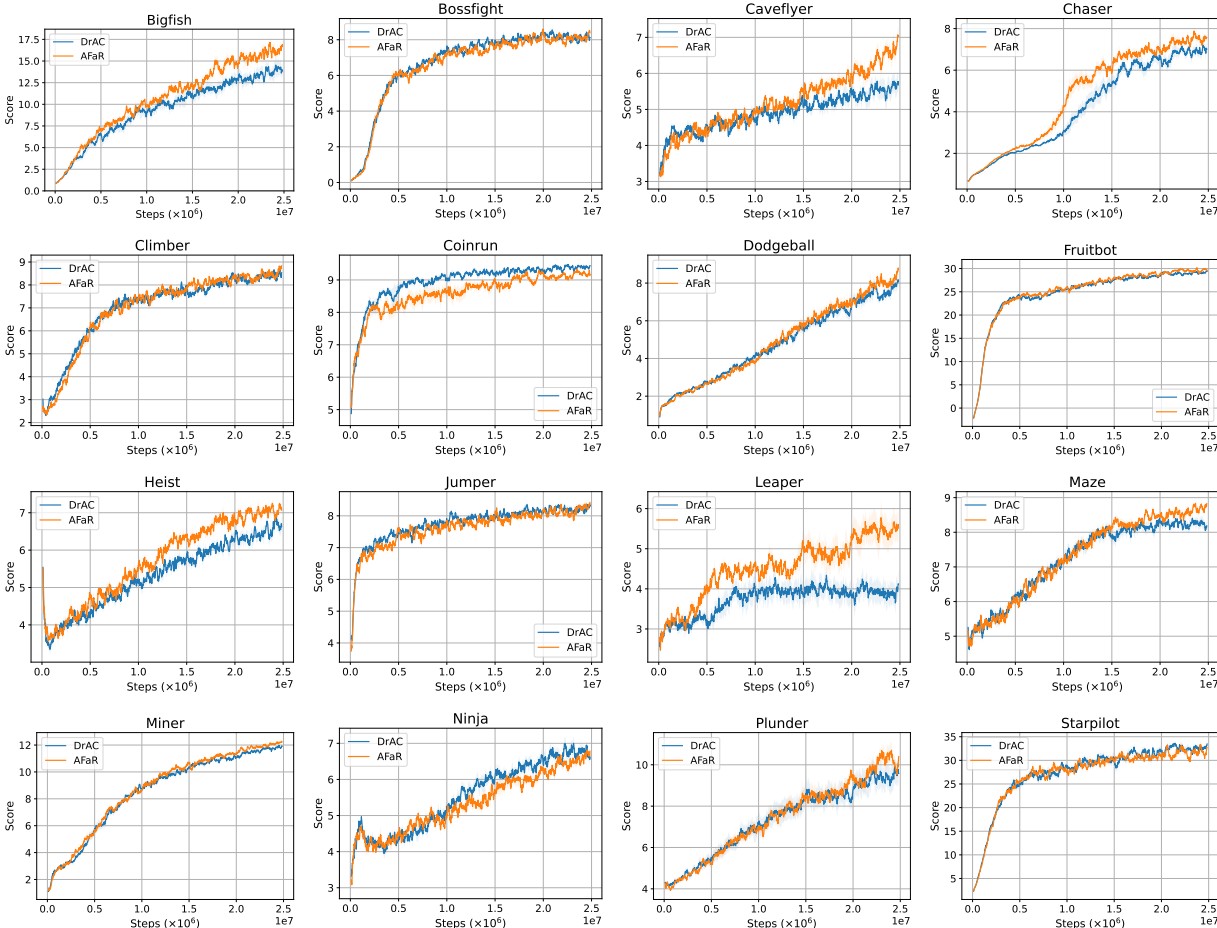

Figure 6: We compare the performance of AFaR with DrAC (as the baseline approach) on the training environments from Procgen Benchmark when training with 200 levels (per environment) for $25M$ steps. The solid curves correspond to the mean performance (score), while the shaded regions represent the standard error (computed over ten runs). There is a substantial performance improvement in 4 environments, and the performance of the proposed approach is close to the baseline approach in the remaining environments. For the performance on the evaluation environments, refer to Figure 7.

In tables 6 and 9, we describe the space of values that we explored for $k$. We picked the best value of $k$ by comparing the performance over 3 runs (seeds) for different values (of $k$). Then we re-ran 10 experiments (seeds) with the best value of $k$ and reported the corresponding results in the paper. The best performing values of $k$ are given in the table 7 for the ProcGen environments and was 4 for the MiniGrid environments. We note the values are very consistent across the implementations, with the $k = 4$ being the most common value. The case where we need more than 4 encoders are environments like DodgeBall where the agent can interact with several "objects" (or factors). We also note that in several environments (like FruitBot or Leaper), the performance across different values of $k$ is quite close.

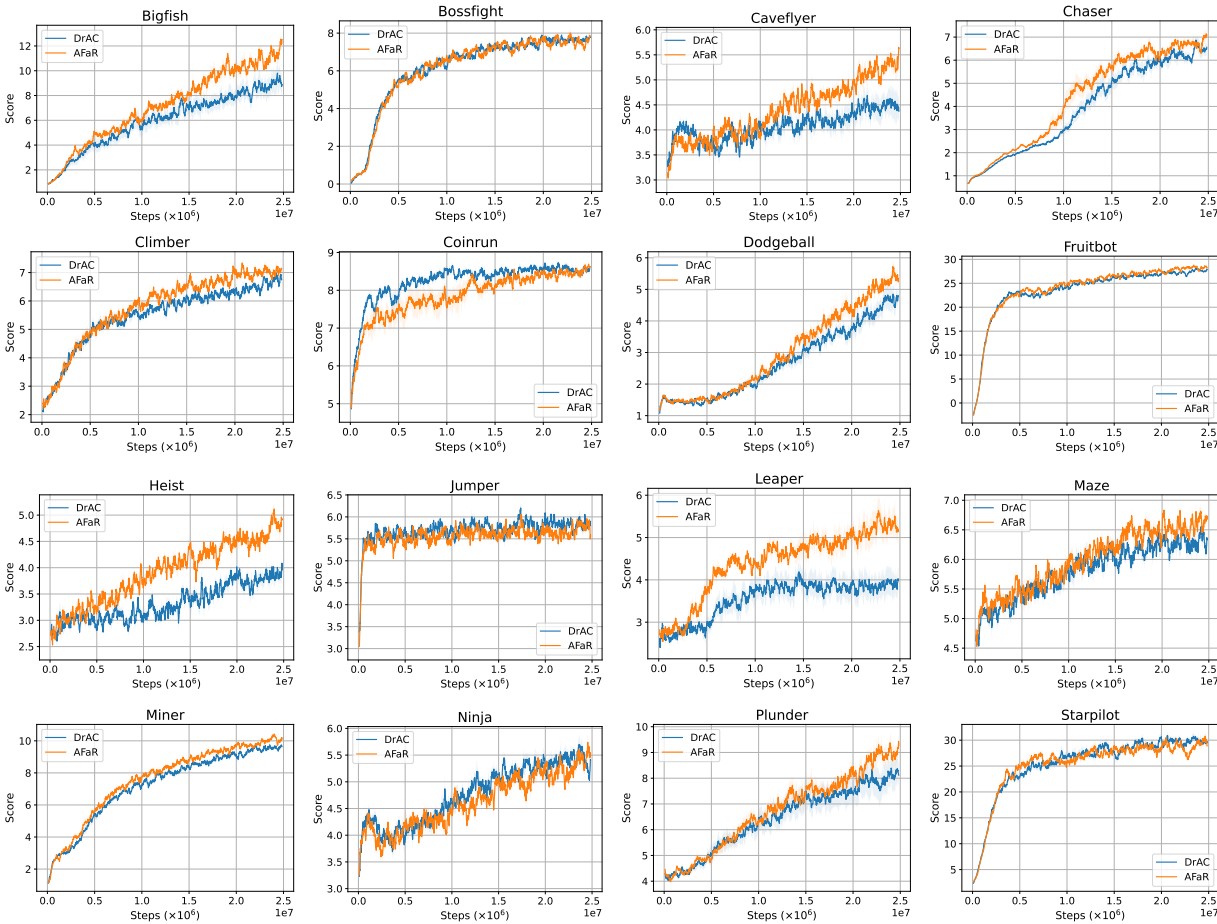

Figure 7: We compare the performance of AFaR with DrAC (as the baseline approach) on the evaluation environments from Procgen Benchmark when training with 200 levels (per environment) for $25M$ steps. The solid curves correspond to the mean performance (score), while the shaded regions represent the standard error (computed over ten runs). There is a substantial performance improvement in 7 environments, and the performance of the proposed approach is close to the baseline approach in the remaining environments. For the performance (on the evaluation environments at the end of the training, refer to Table 1 and for performance on the training environments, refer to Figure 6.

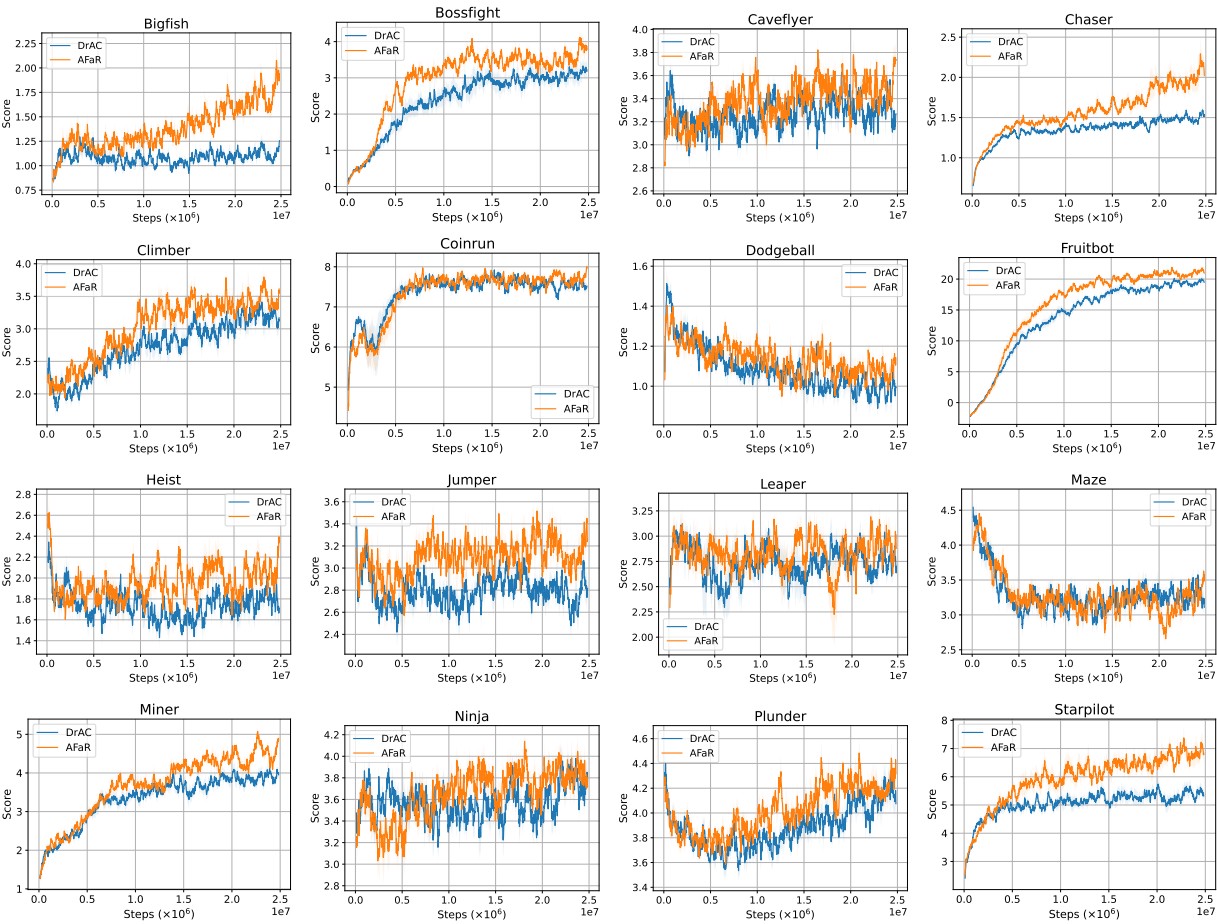

Figure 8: We compare the performance of AFaR with DrAC (as the baseline approach) on the evaluation environments from Procgen Benchmark when training with ten levels (per environment) for $25M$ steps. The solid curves correspond to the mean performance (score), while the shaded regions represent the standard error (computed over ten runs). For corresponding curves when training on 200 levels per environment, refer to Figure 7.

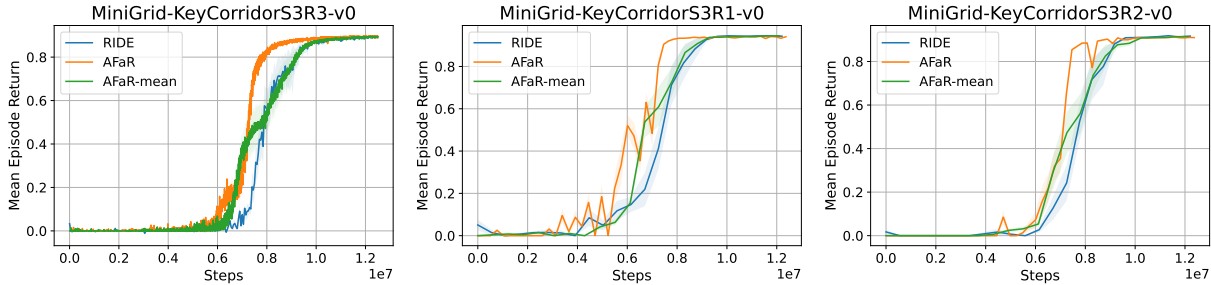

Figure 9: We compare the performance of AFaR with RIDE (baseline approach) on `KeyCorridorS3R3-v0`, `KeyCorridorS3R1-v0` and `KeyCorridorS3R2-v0` environments (left to right) with the number of environment interactions (in millions) along the x-axis and the mean episodic return along the y-axis. We also include an ablation version of the AFaR algorithm where we *disable* the *Attention Module* and assign equal weights to all the factors while aggregating them. We denote this ablation as AFaR-mean. Note that while the agent is evaluated on three environments (`KeyCorridorS3R3-v0`, `KeyCorridorS3R2-v0`, `KeyCorridorS3R1-v0`), it was trained only on the `KeyCorridorS3R3-v0` environment, thus the evaluation on `KeyCorridorS3R2-v0` and `KeyCorridorS3R1-v0` environments is done in a zero-shot manner. For all the three environments, AFaR obtains the best sample efficiency.

## C   Additional Implementation Details

### C.1   Libraries

We use the following open-source libraries:

1. PyTorch Paszke et al. (2019)[7]

2. Hydra Yadan (2019)[8]

3. Numpy Harris et al. (2020)[9]

4. Pandas pandas development team (2020)[10]

5. RIDE Implementation Raileanu & Rocktäschel (2020)[11]

6. DrAC Implementation Raileanu et al. (2020)[12]

7. IDAAC Implementation Raileanu & Fergus (2021)[13]

In case feedforward layers are used to process the output of convolutional encoders, we can additionally vary the size of the intermediate layers in the feedforward networks (corresponding to the different factors). Using variable-sized networks can further help the network to model various factors more effectively.

## D   Testing for statistical significance

We perform a two-tailed, Student's $t$-distribution test Student (1908) under *equal sample sizes, unequal variance* setup (also called Welch's $t$-test). The null hypothesis is: that the mean performance of the two models (AFaR and any baseline) are equal. The significance level ($p$) is set to 0.05.

---

[7] https://pytorch.org/

[8] https://github.com/facebookresearch/hydra

[9] https://numpy.org/

[10] https://pandas.pydata.org/

[11] https://github.com/facebookresearch/impact-driven-exploration

[12] https://github.com/rraileanu/auto-drac

[13] https://github.com/rraileanu/idaac

# E   Baselines

## E.1   Choice of Actor-Critic Methods

In Section 4, we noted that AFaR is a representation learning algorithm, and we need to pair it with a policy optimization algorithm for end-to-end reinforcement learning. We consider three actor-critic algorithms that are shown to obtain state-of-the-art performance in the given environments. For the Procgen environments, we use two baselines: (i) Data-regularized Actor-Critic (DrAC) Raileanu et al. (2020), and (ii) Invariant Decoupled Advantage Actor-Critic (IDAAC) Raileanu & Fergus (2021). The key difference between these two approaches is that IDAAC is the state-of-the-art among methods that use a separate policy and value function, while DrAC is the state-of-the-art among methods that use a shared network for policy and value functions. For MiniGrid environments, we use Rewarding Impact-Driven Exploration (RIDE) Raileanu & Rocktäschel (2020) method, a state-of-the-art method for tasks on the MiniGrid environments.

There are several benefits of using these diverse methods as a baseline for AFaR. These methods focus on different challenges in training RL agents and use different inductive biases and approaches to achieve good performance, using exploration, disjoint policy and value functions, and data augmentation. Moreover, these methods also offer diversity in how the encoders and value functions are trained. For example, IDAAC learns a value function used to provide the learning signal for an advantage function, which is used for training the policy (thus acting as the critic). The advantage function also shares parameters with the actor network as is done in standard actor-critic architectures. i.e. in IDAAC, the actor and the advantage function use the same encoder. Since the advatnage function plays the role of critic, we factorize the advantage function. RIDE uses two sets of encoders – one for training the policy and the other for computing the intrinsic rewards. We factorize only the encoder used for training the policy. Integrating AFaR with diverse baselines and demonstrating that AFaR can improve over these baselines shows that our proposed algorithm is helpful for various actor-critic algorithms.

Next, we describe the different actor-critic algorithms in detail.

DrAC builds on the idea of using data augmentation for improving generalization in RL (Laskin et al., 2020; Srinivas et al., 2020; Kostrikov et al., 2020), and proposes two novel regularization terms for the policy and value function to make the use of data augmentation theoretically sound for actor-critic algorithms. Specifically, given an image transformation $f$, the following two regularization terms are added to the actor-critic loss: $G_\pi = KL[\pi_\theta(a|s), \pi_\theta(a|f(s, \nu)]$ and $G_v = (V_\phi(s) - V_\phi(f(s, \nu)))^2$ where $\nu$ are the parameters of $f(.)$. These additional terms ensure that the learning agent's policy and value functions are invariant to the transformations induced by the augmentations.

MiniGrid environments use the sparse reward setup, making exploration a key challenge. RIDE proposed a novel intrinsic reward where the agent is rewarded for taking actions that affect its learned state representation. Specifically, the intrinsic reward is computed as the $L_2$ norm of the difference between the representation of the consecutive states, i.e. $||\phi(s_{t+1}) - \phi(s_t)||$. Episodic state visitation counts discount this reward to ensure that the agent does not simply go back and forth between the same two states. The overall intrinsic reward is given as: $r_t^i = \frac{||\phi(s_{t+1}) - \phi(s_t)||}{\sqrt{N_{ep}(s_{t+1})}}$, where $N_{ep}(s)$ is the number of time state $s$ is visited in episode $e$. The agent is trained to maximize the weighted sum of the intrinsic reward ($r_t^i$) and the extrinsic reward (obtained from the environment). On the MiniGrid environments, RIDE is reported to improve sample efficiency as compared to approaches like count-based exploration Bellemare et al. (2016), Random Network Distillation (uses prediction error of a random network as the exploration bonus) Burda et al. (2019), Intrinsic Curiosity Module (uses a curiosity based intrinsic reward) Pathak et al. (2017) and IMPALA (an actor-critic approach that uses only external rewards) (Espeholt et al., 2018).

