# OpenReview forum: "Improving Generalization with Approximate Factored Value Functions"
_TMLR — Accepted by TMLR_

### Review · Reviewer_vDzx · 2022-09-28

**Summary Of Contributions:**

This paper introduces a new representation learning method that leverages reward factorization, and shows that this method (combined with baseline actor-critic methods) can improve generalization performance on Procgen and sample efficiency on MiniGrid.

**Requested Changes:**

See weaknesses above.

**Strengths And Weaknesses:**

Strengths
* The paper is in general clearly written.

Major weaknesses:
* The baseline results of DrAC and IDAAC does not match (are noticeably lower than) those reported in their original papers, which may render the comparison unfair.
* Even if only considering the scores in this paper, I do not think the improvement is significant. Besides, I suggest that the authors report the normalized score over all 16 environments and use this new evaluation metric [1].
* Experiments on MiniGrid are limited. Please consider running experiments on other environments, such as MultiRoom and ObstructMaze.
* Some ablation experiments are missing. First, all ablation experiments should be conducted on both MiniGrid and Procgen, as the focuses of these two benchmarks are different. Second, the authors should add experiments with the attention mechanism in Eqn.(2). Third, since the highlight of this paper is reward factorization, it would be better to have ablation experiments on it.

Minor weaknesses:
* Some notations in Section 2.1 are inconsistent or incorrect.
  * In Assumption 2.1 the reward function depends on $s_t$, $a_t$ and $s_{t+1}$, but in Proposition 2.2 it only depends on $s_t$, $a_t$.
  * The expression $r:\mathcal{S}\times\mathcal{A}\to\mathbb{R}$ in Proposition 2 is incorrect. For different factor $i$, $r_i$ is different since the sub state spaces are different. It should be $r_i:\mathcal{S}^i\times\mathcal{A}\to\mathbb{R}$.
  * Regarding the factor-wise value functions (two equations below Eqn.(1)), $\pi$ appears on the left-hand-side but does not appear on the right-hand-side.


[1] Agarwal, Rishabh, et al. "Deep reinforcement learning at the edge of the statistical precipice." Advances in neural information processing systems 34 (2021): 29304-29320.

---

> ### Author Response · Authors · 2022-10-16
> **Thank you for the review!**
>
> We thank the reviewer for their constructive feedback. While we are still working on addressing some parts of the review, we wanted to share our response on the other parts. We will post another response once we have address all the feedback from the reviewer.
>
> > The baseline results of DrAC and IDAAC does not match (are noticeably lower than) those reported in their original papers, which may render the comparison unfair.
>
> We confirm that for all the experiments (including the ones with DrAC and IDAAC), we used the official codes released by the authors [DrAC Official Code](https://github.com/rraileanu/auto-drac) [IDAAC Official Code](https://github.com/rraileanu/idaac). Moreover, the baseline related hyper-parameters for the baseline and baseline + AFaR version were the same.
>
> Regarding DrAC, we noticed that the performance on 4 envs (Caveflyer, Jumper, Leaper and Plunder) is different than the best performance reported in the base line paper. It seems that for these 4 environments, the baseline did not use "crop" as the augmentation to obtain the best results, while we used the "crop" augmentation for all the results as it was the default in the official code. We apologize for this oversight and are re-running results on these 4 environments with other augmentations as well.
>
> Regarding IDAAC, we agree with the reviewer that there are some environments where our reproduced results are worse than the results provided in the baseline paper and a few envs where the results are better than the reproduced results. We are re-running the baseline results to make sure we did not make any mistakes in setting the hyper-parameters. If we do not find any hyper-parameters related issue, we will update the baseline results in our work to match the ones in the published work. If we do find some mistake in using the hyper-parameters, we will re-run the proposed algorithm as well and update both set of results.
>
> > Even if only considering the scores in this paper, I do not think the improvement is significant.
>
> We will address this once we have verified that we used the correct set of hyper-parameters.
>
> > Besides, I suggest that the authors report the normalized score over all 16 environments and use this new evaluation metric [1].
>
> Thank you for the suggestion. We will address this once we have verified that we used the correct set of hyper-parameters.
>
> > Experiments on MiniGrid are limited. Please consider running experiments on other environments, such as MultiRoom and ObstructMaze.
>
> Thank you for the suggestion. We are working on this result.
>
> > First, all ablation experiments should be conducted on both MiniGrid and Procgen, as the focuses of these two benchmarks are different.
>
> Thank you for the suggestion. We are working on this result.
>
> > Second, the authors should add experiments with the attention mechanism in Eqn.(2).
>
> We considered an experiment where the  attention mechanism is replaced by the mean operation. Is there some other ablation that the reviewer has in mind?
>
> > Third, since the highlight of this paper is reward factorization, it would be better to have ablation experiments on it.
>
> We consider ablations where we vary the number of factors. Is there some other ablation that the reviewer has in mind?
>
> > Some notations in Section 2.1 are inconsistent or incorrect.
>
> Thanks for highlighting these. They have been fixed in the updated version.

---

### Review · Reviewer_pydn · 2022-10-07

**Summary Of Contributions:**

The primary focus of this paper was a novel architecture for representations learning based on reward factored MDPs. As I see it there are two main contributions:
1. The novel architecture (AFaR) and describing this architecture in terms of reward factored MDPs.
2. A performance comparison of AFaR with other schemes to improve generalization. Specifically, DrAC and IDAAC in the procgen environment, and RIDE in the MiniGrid-KeyCorridorSxRy environments.

The proposed architecture uses factored states learned implicitly to represent a value function. These states are then passed to the policy network (without gradient connections) and used to learn a policy in tandem with learning the states. The paper sets out to answer three questions related to the generalization performance of the proposed architecture, and some ancillary properties of the architecture (i.e. does it work, sensitivity to the number of factors).

**Requested Changes:**

I've presented a lot of comments, but there are some more serious than others. Below I'll list the most critical suggestions/comments that need to be addressed.

*Need to be addressed:*
- CL-(1, 2, 3, 4)
- EE-(1, 2, 3, 4, 5, 6, 7, 10, 13)

I would like to see all the points addressed or discussed in a rebuttal, but understand there is only so much time.


**Strengths And Weaknesses:**

# Strengths:
- S1. The paper is generally well constructed, and the overall motivation and main line of research is clear to the reader.
- S2. The algorithm description and architecture diagram were useful in understanding the approach proposed.
- S3. The paper attempts to do proper significance testing (using Welch's t-test) for some of the comparisons.


# Weaknesses:

Generally, I like to organize my critiques of a paper based on three criteria: overall clarity of the paper, consistency between the different parts of the paper, well organized and motivated empirical evaluation of the stated research questions, and conclusions which are supported by the provided work.

## Clarity:

Overall, the paper is clear but there are a few doubts on how the algorithm works in detail.

CL-1). Step d of your architecture and line 8 of the algorithm are a bit confusing. For each factored state $z^i$ is there a separate corresponding function $V_i^\pi(z^i)$ to construct the factored value function $V_i^\pi$, or is there a single parameterized function $V^\pi(z^i)$ which constructs each value function $V_i^\pi$? Perhaps a more concrete way of asking is "are the weights for the mapping $V: Z -> \mathbb{R}$ shared for each factored state or different for each factored state"?

CL-2). The second paragraph of section 3 feels a bit too broad, and muddles the motivation for the new architecture. Some questions I think are worth considering if the authors want to motivate their approach through the mean-field principle:
   CL-2.1). Is the factorization of the value function of a un-factorized state into a sum of value functions of factorized states the only possible application of the mean-field principle in this setting?
   CL-2.2). Are we testing the mean-field principle/approximation generally in this setting, or are we testing one particular factorization? If we are testing one particular factorization, I think making the connection to the mean-field principle in the intro/related works would be an interesting addition with a more concrete discussion.

CL-3). You state the hypothesis: "If the factorization is optimized to match the value function, the mean-field approximation may perform well in practice" in section 3. Orthogonal the confusion I have above w/ the connection to the mean-field principle, the hypothesis stated here is imprecise and should be refined into something that is testable. The main issue with the statement is "may perform well in practice". While the authors may have a intuitive understanding of what they mean, this is by no means a testable/communicable hypothesis. Potentially this was written to try and soften the claims the paper makes, but hypotheses can be strong as long as they are testable and the authors test the validity of the hypothesis rather than form a hypothesis to explain their results.

I would suggest the authors try and specify exactly what it means to perform well in practice, or better yet consider what hypothesis would be interesting to pose. Some potentially useful questions:
  - Are you more focused on generalization (which seems to be the focus later in section 5) or generally to perform better than all other approaches?
  - Do you expect the factorization to be beneficial broadly or in certain environments?
  - Do you have some idea of what performing well means? Do the baselines "perform well"?
  - How do we determine if the factorization is or isn't optimized to "match the value function"? Is this only due to the loose conditioning on the training from the out-going objective? Is there a stronger constraint I'm missing? Can we definitively say other networks don't implicitly learn this kind of factorization?

CL-4). It is unclear how RIDE is incorporated into AFaR or how it is changed to make it a baseline. I'm specifically talking about the second paragraph of 5.4.2. This is incredibly unclear, and should be expanded on (at least in the appendix).


## Consistency:

CO-1). The main consistency issues I have are shared in some of the clarity questions above (specifically CL-2, and CL-3). Generally, I don't think positioning the main hypothesis around the mean-field principle and a loose definition of "may perform well in practice" is actually what the authors really mean (and actually suggest later in the paper). This needs to be made consistent.

## Empirical Evaluation:

My main concern with this paper is the hanging questions which are avoided in favor of primarily testing downstream performance. The questions posed seem to be constructed as a consequence of the benchmarking done rather than novel questions posed and then tested. While this is generally not problematic, my issue is the avoidance of a main thrust underlying all of your claims. First, I want to go over the missing research question that needs answered for the other questions and the hypothesis to be tested and discussed. Second, I want to discuss the questions actually posed by the authors, and consider if these are appropriately interesting and if they should be posed differently. Finally, I will discuss some quibbles I have with how the experiments are laid out and some edits.

### Missing Research Question

The main underlying claim the authors use as motivation is that their architecture factorizes the value function in accordance to a linear reward factored MDP. This claim, while maybe intuitive to the authors from their architectural choices, hasn't being evaluated or posed as an empirical question. I think testing this empirically is critical for the claims in the paper to be adequately supported. Below I've listed several recommendations/possible directions which might be interesting to fully analyze the network.

EE-1). An evaluation of the individual value functions (i.e. $V(Z^i_t)$) and how they compare to the total value function.
     - Are all the value functions contributing to the final value function equally? (i.e. are they just learning $\frac{1}{k} V(s_t)$, or something else).
     - If we compare instead to a feed-forward network which learns the value function directly (rather than like AFaR) and the output of a hidden layer is used in the same capacity as $z$.

EE-2). How do the different states $(z^i)$ represent the underlying state? Do they look entirely different?

EE-3). If AFaR is used on an MDP which has a clear reward factoraization will the network learn that factorization? I would really like to see some experiments where we have control over the factorizations, but I'm unaware of some specific examples you could use. Maybe something hand constructed?

### Posed Questions

Next I'll turn to the empirical questions posed in the paper. I'll go in order as they are posed.

EE-4). (i). "Can we use factored representations learned by the AFaR algorithm to train an RL policy?"

This question suffers from some of the issues mentioned in CL-3. Specifically, it being interpreted as "is it possible to use factored representations...". The answer to this is yes, as your architecture uses it. Another way I could interpret this question is "Will the policy learned using only a factored representation be useful?" I think this is the interpretation we are wanting, but the current question doesn't say this clearly.

EE-5). (ii). "Does AFaR improve generalization to new environments where factors vary in novel ways?"

There are two parts of this question that are unclear. What constitutes a new environment and what do you mean by "where factors vary in novel ways?". I think the authors need to clarify what they mean in some discussion


EE-6). (iii). "How robust is AFaR to hyper-parameters like the number of factors that the agent learnings?"

In the empirical section, you only seem to check the number of factors and not other hyper-parameters. So you should focus this question specifically to that hyper-parameter.

### Quibbles

EE-7). A huge quibble I have with how the results are presented are through (what I honestly believe) is a misrepresentation of the actual results shown. If your performance gain is not statistically significant, your null hypothesis (i.e. the means are the same) is true. This means the results (which you've bolded but not added an astrix in your tables) are not improvements over the baselines. There are two changes required, you need to adjust the claims made in the text of the paper and you need to follow what is standard across the field (at least to my knowledge) which is to bold only statistically significant best performers.

EE-8). I'm hesitant to trust the shaded region in Figure 3. If you added all the individual learning curves in the appendix, this would help add evidence to your claim that the lines are significant.

EE-10). Did you only run the results in table 3 with one seed? If so we can't really make any claims wrt

EE-11). It is unclear how the different baselines (and your method) compare on number of tunable parameters. This should be very clearly described and discussed, and if this is controlled for in your empirical evaluation you should mention this more clearly.

EE-12). You included the state-action value function factorization in prop 2.2. I think an empirical evaluation over how this factorization would work in a DQN type network would be really interesting. I think this would more directly support your hypothesis that the specific factorization is helpful for a control algorithm.

EE-13). What are your network architectures specifically? It is unclear given the current state of the paper. For the empirical evaluation in MiniGrid, do you use recurrence to solve the problem? If so how is this put into your networks?


### Supported Conclusions:

I've talked about this quite a bit above. So I'll just refer to the above review for this criteria.

### Misc Questions:

Q-1). In Figure 3, there are some weird results that require explanation (assuming EE-8 is appropriately addressed):
     - Why does the factors set to 4 perform so much worse than those set to 2?
     - What is being learned by each factored value function?
     - It isn't enough to state that "we note that in some cases the choice of k does not make much difference...However, in general, the performance can be improved by tweaking k". I would expect a clear explanation of the behavior given your posed question (discussed in EE-6).

Q-2). Would a MoE network be a good baseline to your approach? Why not use a more basic MoE network as compared to AFaR?

### Misc Comments:

MC-1). I would recommend re-writting the last paragraph of section 7. Currently, it is confusing. I think it is supposed to be focused on future work, but this isn't clear as currently stated.

MC-2). The second paragraph in the introduction is a bit dense, and I'm not sure the cognitive science perspective is adding to the motivation much. It might be worthwhile being further explored in the related work section.

MC-3). If you are referencing Figures 6 and 7 in the main text you should include them in the main paper (you still had a page of space!).

### Minor Edits:
These are edits that are just minor mistakes, and don't factor into my thoughts on the paper.

- The equations in both section 2.2 and appendix A often have hanging parenthesis at the end.

---

> ### Author Response · Authors · 2022-10-16
> **Thank you for the review! 1/2**
>
> We thank the reviewer for their detailed and constructive feedback. While we are still working on addressing some parts of the review, we wanted to share our response on the other parts. We will post another response once we have addressed all the feedback from the reviewer.
>
> > CL1: For each factored state is there a separate corresponding function
>
> On page number 4, we mention that " the neural network (instantiating the value functions) are shared across all factors (shown in component d in Figure 1)"
>
> > CL-2.1). Is the factorization of the value function of a un-factorized state into a sum of value functions of factorized states the only possible application of the mean-field principle in this setting?
>
> It seems there is some confusion regarding the mention of the mean-field principle. The argument of our work is not to "find the best possible way of using the mean field approximation". To answer the more general question of "only possible application of the mean-field principle in this setting", there may be other ways to leverage the mean-field approximation, but they are not the focus of this work.
>
> > CL2: "The second paragraph of section 3 feels a bit too broad, and muddles the motivation for the new architecture." and CL-2.2). "Are we testing the mean-field principle/approximation generally in this setting, or are we testing one particular factorization? If we are testing one particular factorization, I think making the connection to the mean-field principle in the intro/related works would be an interesting addition with a more concrete discussion."
>
> Thanks for the feedback. We have moved this paragraph to the related work and added some more discussion.
>
> > CL-3). the hypothesis stated here is imprecise and should be refined into something that is testable.
>
> Thanks for the feedback. We have re-worded the "may perform well in practice" phrase and mention that we are focusing on generalization and sample complexity.
>
> > CL-4). It is unclear how RIDE is incorporated into AFaR or how it is changed to make it a baseline. I'm specifically talking about the second paragraph of 5.4.2. This is incredibly unclear, and should be expanded on (at least in the appendix).
>
> Thanks for the feedback. We have added the following to the second paragraph of 5.4.2: "Further, we modify the actor-critic components of RIDE as per~\cref{alg:main_algorithm}. RIDE uses a recurrent network for maintaining a history over the observations. We share the weights of the recurrent network across the different factors (with every factor having their own hidden state)." Is there any specific aspect that the reviewer would like us to clarify?
>
> >  EE-1). Are all the value functions contributing to the final value function equally? (i.e. are they just learning
>
> Thanks for suggesting this experiment. We are working to add a plot in the paper where show the output of the different value functions over time (averaged across timesteps and batch)
>
> >  If we compare instead to a feed-forward network which learns the value function directly (rather than like AFaR) and the output of a hidden layer is used in the same capacity as Z
>
> Is the reviewer suggesting that we not factorize the value function? This would be the case with all the baselines that we considered.
>
> > EE-2). How do the different states (zi) represent the underlying state? Do they look entirely different?
>
> Thanks for suggesting this. We are working to add a plot where we show the pairwise MSE distance between the states after training. (averaged across timesteps and batch).
>
> > EE-3). If AFaR is used on an MDP which has a clear reward factorization will the network learn that factorization? I would really like to see some experiments where we have control over the factorizations, but I'm unaware of some specific examples you could use. Maybe something hand constructed?
>
> Thanks for suggesting this. We will try to work on this within the rebuttal period.
>
> > E-4). (i). "Can we use factored representations learned by the AFaR algorithm to train an RL policy?"
> This question suffers from some of the issues mentioned in CL-3. Specifically, it being interpreted as "is it possible to use factored representations...". The answer to this is yes, as your architecture uses it. Another way I could interpret this question is "Will the policy learned using only a factored representation be useful?" I think this is the interpretation we are wanting, but the current question doesn't say this clearly.
>
> Thanks for the feedback. We have reworded the question

---

> > ### Author Response · Authors · 2022-10-16
> > **Thank you for the review! 2/2**
> >
> > > EE-5). (ii). "Does AFaR improve generalization to new environments where factors vary in novel ways?"
> > There are two parts of this question that are unclear. What constitutes a new environment and what do you mean by "where factors vary in novel ways?". I think the authors need to clarify what they mean in some discussion
> >
> > Thanks for the feedback. We have added the following to the main paper: "Here, ``new environments'' refer to environments with different layouts (or sizes) or of a different level of difficulty than the original environments. ``Factors vary in novel way'' refer to change in properties of the factors that the agent saw during training. For example, in the BigFish environment (from ProcGen), the number/size/shape of fishes varies across the environments. Similarly, in the MiniGrid environment, the number of doors and their types (i.e. locked or unlocked) varies across environments. "
> >
> > > EE-6). (iii). "How robust is AFaR to hyper-parameters like the number of factors that the agent learnings?"
> > In the empirical section, you only seem to check the number of factors and not other hyper-parameters. So you should focus this question specifically to that hyper-parameter.
> >
> > Thanks for the feedback. We have reworded the question
> >
> > EE-7). A huge quibble  ...  bold only statistically significant best performers.
> >
> > Thanks for the feedback. We have reworded the presentation of the results and highlight only the statistically significant results.
> >
> > > EE-8). I'm hesitant to trust the shaded region in Figure 3. If you added all the individual learning curves in the appendix, this would help add evidence to your claim that the lines are significant.
> >
> > Thanks for the feedback. We will add the individual curves to the paper. We also highlight that in Figure 3, we do not claim that the presented results are significant.
> >
> > > EE-10). Did you only run the results in table 3 with one seed? If so we can't really make any claims wrt
> >
> > All the experiments (including the ones in Table 3) are run with 10 seeds. We compute the mean of the success rates (across 10 seeds) and report the number of environment steps when this mean reaches the 95% success threshold.
> >
> > > EE-11). It is unclear how the different baselines (and your method) compare on number of tunable parameters. This should be very clearly described and discussed, and if this is controlled for in your empirical evaluation you should mention this more clearly.
> >
> > Thank you for the feedback. We will include a discussion on the addition of number of parameters because of using the AFaR approach and will update here once this change has been done in the paper.
> >
> > > EE-12). You included the state-action value function factorization in prop 2.2. I think an empirical evaluation over how this factorization would work in a DQN type network would be really interesting. I think this would more directly support your hypothesis that the specific factorization is helpful for a control algorithm.
> >
> > Thanks for suggesting this. We will try to work on this within the rebuttal period.
> >
> > > EE-13). What are your network architectures specifically? It is unclear given the current state of the paper. For the empirical evaluation in MiniGrid, do you use recurrence to solve the problem? If so how is this put into your networks?
> >
> > The network architectures depend on the baseline that we integrate with. For the MiniGrid experiments, we do use recurrence, as was used in the baseline algorithm. We share the weights of the recurrent network across the different factors (with every factor having their own hidden state). We have updated this in the paper.
> >
> >
> > > Q-1). In Figure 3, ...  our posed question (discussed in EE-6).
> >
> > Thank you for the question. We are working to address this.
> >
> > > Would a MoE network be a good baseline to your approach? Why not use a more basic MoE network as compared to AFaR?
> >
> > AFaR is a MoE network. Does the reviewer have a specific MoE architecture in mind ?
> >
> > > MC-1). I would recommend re-writting ... currently stated.
> >
> > Thanks for the feedback. We will reword the last paragraph and update here once this change is done in the paper.
> >
> > > MC-2). The second paragraph in the introduction is a bit dense, and I'm not sure the cognitive science perspective is adding to the motivation much. It might be worthwhile being further explored in the related work section.
> >
> > Thanks for the feedback. We will try re-wording the motivation and update here once this change is done in the paper.
> >
> > > MC-3). If you are referencing Figures 6 and 7 in the main text you should include them in the main paper (you still had a page of space!).
> >
> > Thanks for the feedback. We will move them in the main paper and update here once this change is done in the paper.
> >
> > > Minor Edits:
> > These are edits that are just minor mistakes, and don't factor into my thoughts on the paper.
> > The equations in both section 2.2 and appendix A often have hanging parenthesis at the end.
> >
> > Thanks for the feedback. These have been corrected.

---

### Review · Reviewer_fNHM · 2022-10-08

**Summary Of Contributions:**

This paper studies the important problem of learning factored representations in RL; rather than assuming factored MDPs, this paper attempts to learn a factored approximation of the underlying MDP. They propose an attention-based architecture that learns factored value functions that when summed together, yield the agent’s current value function $\hat{V*}$. The learned factors in conjunction with their corresponding values are used as the input representation to a policy network, which is trained using a classic actor loss. This modular architecture outperforms baseline actor-critic algorithms on procgen and minigrid.

**Broader Impact Concerns:**

I dont have any concerns about the broader impact of this work

**Requested Changes:**

- Ablations: actor-critic baseline with the same number of parameters as AFAR and actor-critic with attention. Such an ablation will strengthen the paper.
- A discussion about Hybrid Reward Architecture (HRA) from Van Seijen et al and how it fits into the factored MDP landscape; this will also strengthen the paper.
- Some experiments (maybe qualitative) that showcase that the proposed architecture is learning meaningful factors that wouldn’t automatically be learned by a vanilla actor-critic agent; I think such experiments will significantly improve the paper

**Strengths And Weaknesses:**

Strengths:
- paper tackles an important and yet understudied problem in RL
- paper is easy to follow; technical contributions are laid out clearly
- Evaluation is thorough - algorithm is tested on a range of challenging benchmark problems and it consistently seems to do better than its baseline counterparts.

Weaknesses:
- It is not clear to me how we can be confident that the proposed architecture is actually learning a factored representation.
- can the aggregation function on the actor side be summarized as scaled dot-product attention? Or is different in a way that I am missing?
- The discussion about successor features seemed a little weak to me - are there any reasons why we cannot use successor features in Procgen or Minigrid in the same way that we are using the proposed algorithm?

---

> ### Author Response · Authors · 2022-10-16
> **Thank you for the review**
>
> We thank the reviewer for their constructive feedback. While we are still working on addressing some parts of the review, we wanted to share our response on the other parts. We will post another response once we have address all the feedback from the reviewer.
>
> > It is not clear to me how we can be confident that the proposed architecture is actually learning a factored representation.
>
> It is quite difficult to "ensure" that learnt representations are indeed factored. One common failure mode (for methods that try to learn factored representations) is that all the factors become "identical". In that case, learning the attention scores is same as aggregation using the mean operation. As shown in Table 3, AFaR outperforms the case when mean operation is used for aggregation. A second common failure case is when the factors "collapse", i.e. only one factor learns the representation and the other factors learn a degenerate representation. In that case, the attention score, corresponding to the active factor, would be 1 and the attention scores corresponding to the other factors would be 0. And the entropy for the distribution (of attention scores) would be 0. We are working to add a figure where we show the entropy for the distribution of attention scores. We will make an edit once that figure is in the paper.
>
> Another related aspect is that of interpretability, where we interpret if the learned representations can be "mapped" to factors in the environment. The most common way of doing this is to learn a pixel decoder that decodes the factors into images. We did not use a reconstruction loss in our work though future work could consider using approaches like [Slot Encoders](https://arxiv.org/abs/2006.15055) to learn "interpretable" factors.
>
> > can the aggregation function on the actor side be summarized as scaled dot-product attention? Or is different in a way that I am missing?
>
> Common scaled dot-product attention based formulations (https://paperswithcode.com/method/scaled) rely on 3 components - Key, Query and Value. In our case, while we do have the notion of Value (i.e. the representation of the factors), we do not have a clear notion of Keys and Queries. Further, dot-product attention relies on the dot-product operation to compute the attention values while we use a neural network. So while the aggregation operation does have some similarities with the scaled dot-product attention operation, it is not doing the exact same thing.
>
> > The discussion about successor features seemed a little weak to me - are there any reasons why we cannot use successor features in Procgen or Minigrid in the same way that we are using the proposed algorithm?
>
> Successor features are commonly used for transfer across different reward functions. In the case of the MiniGrid and Procgen, the reward function is the same across the environments.
>
> > Ablations: actor-critic baseline with the same number of parameters as AFAR and actor-critic with attention. Such an ablation will strengthen the paper.
>
> Using AFaR does not increase the size of the network significantly. The policy and the baseline networks do not change at all. We change the final layer of the CNN encoders, to scale the number of output channels and introduce a two-layer attention network  with 128 hidden dimensions. In the paper, we have included an ablation (AFaR mean) which uses the larger CNN encoder (but without the attention network).
>
> > A discussion about Hybrid Reward Architecture (HRA) from Van Seijen et al and how it fits into the factored MDP landscape; this will also strengthen the paper.
>
> Thank you for the suggestion. We are working on it.
>
> > Some experiments (maybe qualitative) that showcase that the proposed architecture is learning meaningful factors that wouldn’t automatically be learned by a vanilla actor-critic agent; I think such experiments will significantly improve the paper
>
> Thank you for the suggestion. We are working on it.

---

> > ### Author Response · Authors · 2022-10-24
> > **Update 1**
> >
> > Dear reviewer. We are sharing another update on your previous feedback:
> >
> > 1. The details about the change in the number of parameters (due to use of AFaR) has been added to the paper in Section 5.2 The number of parameters increase by 5 to 10% with addition of new factors (depending on the baseline).
> >
> > 2. We have added new results on the multiroom environments (from the minigrid environments). These environments are larger than the existing key-door environments. Like the previous results on the minigrid environments, AFaR obtains improved sample efficiency on the training and the evaluation tasks.
> >
> > 3. We have added the AFaR-mean baseline for the ProcGen environment (corresponding to the IDAAC baseline).
> >
> > 4. We have improved the performance for both the IDAAC baseline and AFaR by using the best hyper-parameters (per environments) from the IDAAC baseline.

---

> > > ### Comment · Reviewer_fNHM · 2022-10-25
> > > **Follow up**
> > >
> > > Thank you for including this discussion. Unless I am missing something, I don't think that you have provided an ablation with the number of parameters - adding a new "factor" increases the number of parameters in your neural network, I was suggesting an ablation where the baseline has the same number of parameters as the factored agent (to make it a fair comparison). Furthermore, I was suggesting an "attention baseline" where you add attention to the architectures of the baseline agents without making any assumptions about factoredness (which is what I think your contribution is). Apologies if you have already included this and I missed it somehow.

---

> > > > ### Author Response · Authors · 2022-10-25
> > > > **Reply to Follow up**
> > > >
> > > > Thank you for the follow up question.
> > > >
> > > > AFaR-mean has nearly the same number of parameters as the AFaR method. "Nearly" because it does not use the attention network (which has about 256*k or 1204*k parameters, which is negligible compared to the number of parameters the RL network has). If it helps, we can also re-run the baseline experiments to have more parameters in the encoder though it may not improve the performance as the RL network architectures are designed for the baseline methods and not AFaR method.

---

> > > > > ### Comment · Reviewer_fNHM · 2022-10-29
> > > > > **Number of parameters**
> > > > >
> > > > > That's a fair point - no need to re-run the baseline experiments if you are confident that the increase in # parameters is small and insignificant.

---

### Decision · Action_Editors · 2022-11-07

**Recommendation:** Accept with minor revision

**Comment:**

This paper is easy to read and proposes a novel and useful idea. Automatically discovering factorizations is a useful direction and though this paper does not yet give much insight into what factorizations emerge and when this approach might be useful, I believe it will help others continue down this direction.

The decision is Accept with Minor Revisions because I agree with the reviewers that there remains an open question about exactly what is providing performance differences with AFaR. The suggestion to only use attention, without factorization, should be feasible, as this is just a standard MOE approach without explicit critic updates. If that is not a possible ablation, then this could be explained in the work.

A few other changes are also important.
1. The authors stated: "We are working to add a plot in the paper where show the output of the different value functions over time (averaged across timesteps and batch)". This would help understand the learned factorization.
2. The table only seems to bold items if AFaR is better. But typically bolding is used to show which method was statistically significantly better. You should also bold if other method were best. Note if you are willing to add more runs (at least in a few settings), then you could get more clear results.
3. It would be useful to explain for the experiment what factorizations might emerge.
4. I actually could not understand exactly what was meant about learning the factorizations only with the critic, and attention for the policy part. A more explicit algorithm could clarify this, or an objective function could be provided in the section introducing the framework.

Not necessary changes below but just two comments that might be useful for you:
First, the sample efficiency improvements seem to be more important here than final performance results. I wonder if more of the results should focus on this aspect, as again the improvement from AFaR is not that large in terms of final performance. Adding this inductive bias should only improve efficiency, and so that is the natural thing to report.

Second, such a reward factorization has been previously considered for Exogenous MDPs. It only factors into two parts of the state, so it by no means accomplishes what is done here. But, they also note the factorization of the value function. It could be useful to look at and cite this work: "Discovering and Removing Exogenous State Variables and Rewards for Reinforcement Learning", Dietterich et al.

**Audience:**

I found this paper interesting, as did several of the reviewers.

**Claims And Evidence:**

The paper is clearly written and thorough. The biggest omission in the work is better investigating exactly what is learned in the factorization, as well as one ablation experiment to assess the role of attention. It is understandable that the authors were more interested in seeing performance improvements, rather than carefully investigating and interpreting the factorizations learned by the approach. One could argue that it is useful to see overall results, before diving more deeply into the method. This is particularly true if there is a big gain, warranting a deeper study. In this work, the performance improvement were actually quite small though, and likely it would be more impactful to see what is being learned to better understand the impact the approach could have in other settings. I encourage the authors to keep this in mind during their minor revision.

---

> ### Author Response · Authors · 2022-12-13
> **Thanks for the feedback!**
>
> Dear Action Editor
>
> We are thankful for your feedback for improving the paper and have incorporated the feedback as follows:
>
> > The authors stated: "We are working to add a plot in the paper where show the output of the different value functions over time (averaged across timesteps and batch)". This would help understand the learned factorization.
>
> We have added the plot as Figure 4
>
> > "The suggestion to only use attention, without factorization, should be feasible, as this is just a standard MOE approach without explicit critic updates. If that is not a possible ablation, then this could be explained in the work."
>
> Thanks for recommending this baseline. We have added these results for the ProcGen environment in Table 2 and for MiniGrid environments in Table 3 (referred as Factoredized Actor)
>
> > "The table only seems to bold items if AFaR is better."
>
> We clarify that we bold the column corresponding to the best performing algorithm, if it is significantly better than the IDAAC baseline. If the IDAAC baseline is the best performing baseline, we bold the entry of it is significantly better than the AFaR approach. We have clarified it in the caption of the table.
>
> > "It would be useful to explain for the experiment what factorizations might emerge."
>
> We have added some examples for the factorization that the agent could learn.
>
> > "factorizations only with the critic"
>
> The (factored) representations are learnt using the critic loss as explained in section 3.1 and Figure 1. We have further clarified this in Algorithm 1.  The actor "chooses" (assigns weights) to factors using an "attention module" and "aggregation module". These modules are trained using the actor loss as explained in Section 4.1 and Figure 1. We have further clarified this in Algorithm 1.
>
> > It could be useful to look at and cite this work: "Discovering and Removing Exogenous State Variables and Rewards for Reinforcement Learning", Dietterich et al.
>
> Thanks for surfacing this useful reference. We have added this to the related work.
>
> Edit on 13th January, 2023: Formatting changes